# A BAYESIAN FRAMEWORK FOR CLUSTERED FEDERATED LEARNING

## ABSTRACT

One of the main challenges of federated learning (FL) is handling non-independent and identically distributed (non-IID) client data, which may occur in practice due to unbalanced datasets and use of different data sources across clients. Knowledge sharing and model personalization are key strategies for addressing this issue. Clustered federated learning is a class of FL methods that groups clients that observe similarly distributed data into clusters, such that every client is typically associated with one data distribution and participates in training a model for that distribution along their cluster peers. In this paper, we present a unified Bayesian framework for clustered FL which optimally associates clients to clusters. Then we propose several practical algorithms to handle the, otherwise growing, data associations in a way that trades off performance and computational complexity. This work provides insights on client-cluster associations and enables client knowledge sharing in new ways. For instance, the proposed framework circumvents the need for unique client-cluster associations, which is seen to increase the performance of the resulting models in a variety of experiments.

## 1 INTRODUCTION

Federated learning (FL) is a distributed machine learning approach that enables model training on decentralized data located on user devices like phones or tablets. FL allows collaborative model training without data sharing across clients, thus preserving their privacy (McMahan et al., 2017). FL has been applied to computer vision (Shenaj et al., 2023; Liu et al., 2020), smart cities (Zheng et al., 2022; Khan et al., 2021; Park et al., 2022), or threat detection (Wu et al., 2023), among other pervasive applications (Rieke et al., 2020). However, FL faces significant challenges in handling non-independent and identically distributed (non-IID) data, where clients have unbalanced and statistically heterogeneous data distributions (Kairouz et al., 2021; Li et al., 2020; 2022). This violates common IID assumptions made in machine learning and leads to poor model performance.

To overcome non-IID challenges, recent works explored personalizing models to each client while still sharing knowledge between clients with similar distributions (Tan et al., 2022; Huang et al., 2021; Wu et al., 2021). One such approach is clustered federated learning (CFL), which groups clients by similarity and associates each client with a model, which is trained based on the data of clients on the same cluster (Ma et al., 2022; Ghosh et al., 2020; Long et al., 2022). It usually performs better in non-IID data, with clustered clients sharing similar data distribution. However, fundamental questions remain open on how to optimize client-cluster associations and inter-client knowledge sharing under non-IID data. This work aims at addressing two identified problems of current CFL schemes. **Problem 1**: the clients are clustered into nonoverlapping clusters, such that the knowledge of each participating client is exploited by only one cluster during each round, which results in an inefficient utilization of the local information that could contributed to the training of multiple clusters instead. **Problem 2**: there is a lack of a unified theory to describe the information sharing among clients and their contribution to training multiple models.

In this work, we propose a new Bayesian framework that formalizes CFL as a Bayesian data association problem. The global model shared by the server is treated as a mixture distribution, where each component corresponds to an association hypothesis. Instead of clustering clients and associating their data to a given mode, the conceptual solution keeps track of all possible client-cluster associations through probabilistic data association. This results in a principled theory to model both

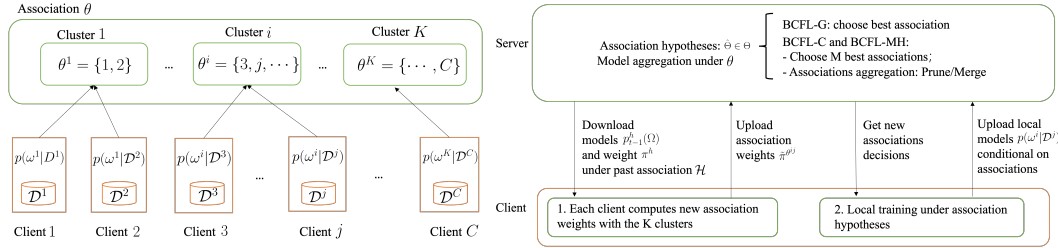

Figure 1: An example (left) association relation between clients and clusters; and details (right) of the operations and communications at the server and client communicate for BCFL.

client-cluster and client-client relations in CFL offering theoretical insights into how to optimize CFL under non-IID settings, as well as connections to existing CFL methods. However, the optimal solution is generally intractable as the number of communication rounds increase due to the quickly growing number of data associations. To address this challenge, the paper also provides three practical algorithms, each leveraging different approximations related to which association hypotheses to keep at each Bayesian recursive update. A variety of experiments are discussed, including both feature-skew and label-skew non-IID situations, showing the systematic superiority of the proposed methods under the general Bayesian CFL (BCFL) framework introduced in this paper.

## 2 RELATED WORKS

Many approaches try to tackle the non-IID issue in FL. Personalized FL has attracted much attention, it customizes models to each client's local data distribution. There are several ways to conduct customization. Local fine-tuning (Ben-David et al., 2010; Wang et al., 2019), meta-learning (Fallah et al., 2020; Jiang et al., 2019), transfer learning (Li & Wang, 2019), model mixture methods (Deng et al., 2020), and pair-wise collaboration method (Huang et al., 2021). However, these methods focus on client-level that does not consider any cluster structure, such that clients with similar backgrounds or distribution are very likely to make similar decisions. Therefore, CFL is proposed as an alternative solution, which provides a middle ground by grouping similar clients and allowing each to associate with a model trained on its cluster distribution (Mansour et al., 2020; Briggs et al., 2020; Sattler et al., 2020b; Shlezinger et al., 2020). This balances personalization with knowledge transfer between related clients. While promising, optimally associating clients to clusters and enabling inter-client learning remains an open area, which this paper addresses. Some of the existing works group the clients by model distance (Long et al., 2022; Ma et al., 2022) and gradient similarity (Sattler et al., 2020a; Duan et al., 2020). Other works utilize the training loss to assign a client to the cluster with the lowest loss (Ghosh et al., 2020; Mansour et al., 2020). However, these clustered FL methods do not effectively exploit similarities between different clusters. Some other works tackle this problem by relaxing the assumption that each client can only be associated with one data distribution, called Soft Clustered FL (Ruan & Joe-Wong, 2022; Li et al., 2021). While those works made substantial progress in different CFL directions, there is a lack of a unifying theory. This article provides a Bayesian interpretation of CFL, where client-cluster assignments are modeled using data association theory (Lee et al., 2014; de Waard et al., 2008). This principled approach enables the design of practical solutions for CFL, some of which have interpretations in connection to the existing works.

Compared to FL, Bayesian FL enhances FL by leveraging the benefits of Bayesian inference. By integrating prior knowledge and inferring parameter distributions, this approach effectively captures the intrinsic statistical heterogeneity of FL models, which facilitates the quantification of uncertainties and model dynamics. Consequently, it fosters the development of more robust and interpretable federated models (Cao et al., 2023). While the majority of existing Bayesian FL research has concentrated on Bayesian training employing Variational Inference (Corinzia et al., 2019; Kassab & Simeone, 2022), Laplace's approximation (Liu et al., 2021), or Bayesian model aggregation (Wu et al., 2022). However, papers combining Bayesian FL with data association mechanisms are notably absent, especially in clustered FL. Addressing this gap is the primary contribution of our paper.

# 3 OPTIMAL BAYESIAN SOLUTION FOR CLUSTERED FEDERATED LEARNING

We envisage (see Figure 1) a FL system with $C$ clients, some of which observe data drawn from similar distributions. The server aggregates local model updates in a way that generates multiple models (each corresponding to different client clustering associations) which are then shared to the clients for further local training. The proposed Bayesian CFL (BCFL) approach does not know the client associations beforehand, instead these are learned leveraging probabilistic data association. The remainder of the section presents the BCFL framework, featuring an optimal solution that accounts for all possible associations; then we discuss a recursive update version where at each FL communication round the model updates only require local updates from the current dataset; finally, we show that such conceptual solution is generally intractable due to the growing number of associations, which motivates the approximations and practical algorithms proposed in Section 4. For convenience, we describe the notation conventions in Appendix A.

## 3.1 BAYESIAN CLUSTERED FEDERATED LEARNING (BCFL) FRAMEWORK

Following a Bayesian paradigm, our ultimate goal is to obtain the posterior distribution $p(\Omega|\mathcal{D})$ of the set of parameters for the $K$ clusters of the server model, $\Omega \triangleq \{\omega^1, \ldots, \omega^K\}$, given the local data, $\mathcal{D} \triangleq \{\mathcal{D}^1, \ldots, \mathcal{D}^C\}$, of all $C$ clients. Furthermore, within the clustered FL philosophy, we aim to associate clients with each of the $K$ clusters (or models). Thus, let $\theta = \{\theta^1, \ldots, \theta^K\} \in \Theta$ be the set of $K$ associations between clients and clusters, where $\theta^i$ is a random finite set (RFS) containing the indexes of all clients associated to cluster $i$, and $\Theta$ be the set of all possible client-cluster associations. Here, $\theta^i$ is a RFS since its cardinality $|\theta^i|$ and its index content are random (Mahler, 2007; Daley et al., 2003). With these definitions, we can expand the posterior distribution using Bayes rule and marginalization of the hypotheses as

$$p(\Omega|\mathcal{D}) \propto p(\mathcal{D}|\Omega)p(\Omega) = \sum_{\theta \in \Theta} p(\mathcal{D}, \theta|\Omega)p(\Omega) \tag{1}$$

where $p(\Omega)$ is the model shared by the server, which is to be updated to compute $p(\Omega|\mathcal{D})$ based on all available data from the $C$ clients.

Besides the complexity associated with the cardinality of $\Omega$, we also aim at a posterior expression that can be evaluated in a distributed manner by fusing local posterior updates from the clients. Alternatively, and without further approximations, we aim to express equation 1 as a mixture distribution where each mode in the mixture represents different client-cluster association hypotheses and which can be (recursively) updated in a more manageable way. More precisely, we target a posterior characterization of the form

$$p(\Omega|\mathcal{D}) = \sum_{\theta \in \Theta} \pi^\theta p^\theta(\Omega) , \tag{2}$$

where

$$\pi^\theta = \frac{\tilde{\pi}^\theta}{\sum_{\theta \in \Theta} \tilde{\pi}^\theta} \qquad \text{where} \qquad \tilde{\pi}^\theta = \int p(\mathcal{D}, \theta|\Omega)p(\Omega)d\Omega \tag{3}$$

$$p^\theta(\Omega) = p(\mathcal{D}, \theta|\Omega)p(\Omega)/\tilde{\pi}^\theta . \tag{4}$$

Notice that an equivalent representation of the posterior is $p(\Omega|\mathcal{D}) = \sum_{\theta \in \Theta} p(\Omega|\mathcal{D}, \theta)\mathbb{P}[\theta|\mathcal{D}]$, which provides an interpretation for the terms in equation 2 as 1) the weights $\pi^\theta$ represent the probability of a data association $\theta$ given available data, $\pi^\theta = \mathbb{P}[\theta|\mathcal{D}]$; and 2) the mixing distributions correspond to the posterior updates given the $\theta$ association, $p^\theta(\Omega) = p(\Omega|\mathcal{D}, \theta)$.

We shall see that, under the assumptions that the clusters are mutually independent (i.e., $p(\Omega) = \prod_i p(\omega^i)$) and that the local datasets are conditionally independent ($\mathcal{D}^i \perp \mathcal{D}^{i'}$, for $i \neq i'$) given $\Omega$ and $\theta$, the mixing distributions in equation 4 can be expressed as

$$p^\theta(\Omega) \propto p(\mathcal{D}^{\theta^1}, \ldots, \mathcal{D}^{\theta^K}, \theta|\Omega)p(\Omega) = \prod_{i=1}^K \underbrace{p(\mathcal{D}^{\theta^i}|\omega^i)p(\omega^i)}_{\propto p^{\theta^i}(\omega^i)} \tag{5}$$

and the weights in equation 3 can be similarly manipulated as

$$\pi^\theta \propto \int \prod_{i=1}^K p(\mathcal{D}^{\theta^i}|\omega^i)p(\omega^i)d\omega^1 \cdots d\omega^K = \prod_{i=1}^K \overbrace{\int p(\mathcal{D}^{\theta^i}|\omega^i)p(\omega^i)d\omega^i}^{\tilde{\pi}^{\theta^i}} = \prod_{i=1}^K \tilde{\pi}^{\theta^i} . \tag{6}$$

It can be observed from equation 5 and 6 that these quantities can be factorized, and thus computed, for each cluster. As a result, the target posterior in equation 2 can be written as

$$p(\Omega|\mathcal{D}) = \sum_{\theta\in\Theta} \pi^\theta p^\theta(\Omega) = \sum_{\theta\in\Theta} \prod_{i=1}^{K} \pi^{\theta^i} p^{\theta^i}(\omega^i) \,. \tag{7}$$

where $\pi^{\theta^i} = \tilde{\pi}^{\theta^i} / \sum_{\theta\in\Theta} \prod_{i=1}^{K} \tilde{\pi}^{\theta^i}$. Note that the normalization required to compute $\pi^{\theta^i}$ must be computed at the server. Finally, further manipulations to the terms in equation 7 can be performed in order to show that both the mixing distribution and the weights, under an association hypothesis $\theta$, can be obtained from local posterior updates in a decentralized manner. First, following the methodology in Wu et al. (2022) we obtain that

$$p^{\theta^i}(\omega^i) \propto \prod_{j\in\theta^i} p(\mathcal{D}^j|\omega^i)p(\omega^i) \propto \prod_{j\in\theta^i} p(\omega^i|\mathcal{D}^j) \,, \tag{8}$$

where the product is over all clients associated with the $i$-th cluster under the current association hypothesis. Therefore, $p(\omega^i|\mathcal{D}^j)$ denotes the local posterior update of the parameters of the $i$-th cluster given the data from the $j$-th client, which can indeed be computed locally. Secondly, we can similarly see that under certain approximations, the weights $\pi^\theta$ can be evaluated as product of locally computed weights $\pi^{\theta^i}$ for a specific data association hypothesis $\theta$. Unfortunately, integral and product in $\pi^{\theta^i}$ cannot be swapped in general, thus we propose a numerical integration approximation that enables the desired factorization as

$$\tilde{\pi}^{\theta^i} = \int \prod_{j\in\theta^i} p(\mathcal{D}^j|\omega^i)p(\omega^i)d\omega^i \approx \sum_{\ell=1}^{N} \alpha_\ell \prod_{j\in\theta^i} \overbrace{p(\mathcal{D}^j|\omega_\ell^i)}^{\tilde{\pi}_{\ell,j}^{\theta^i}} = \sum_{\ell=1}^{N} \alpha_\ell \prod_{j\in\theta^i} \tilde{\pi}_{\ell,j}^{\theta^i} \,, \tag{9}$$

where $\alpha_\ell$ is the weight associated to $\omega_\ell^i$, a sample from $p(\omega^i)$ which can be drawn in different ways. For instance, one could randomly generated those $N$ samples using importance sampling, or the samples could we deterministically generated as done when using sigma-point integration such as Gauss-Hermite rules. An even simpler, yet effective approach is to consider $N = 1$ such that the sample is directly the expected value of $\omega^i$, in which case $\omega_1^i = \mathbb{E}_{p(\omega^i)}\{\omega^i\}$ and we write $\tilde{\pi}^{\theta^i} = \prod_{j\in\theta^i} \tilde{\pi}_j^{\theta^i}$ with $\pi_j^{\theta^i} = p(\mathcal{D}^j|\omega_1^i) \propto \tilde{\pi}_j^{\theta^i}$. Notice that in all cases, weights can be computed locally, under $M$ samples of the distribution $p(\omega^i)$. Additionally, we used the aforementioned conditional independence assumptions and we defined $\pi_j^{\theta^i}$ as the probability of associating client $j$ data to the $i$-th cluster, which again can be computed locally except for the normalization constant $\sum_{\theta\in\Theta} \prod_{i=1}^{K} \tilde{\pi}^{\theta^i}$ which requires all local $\tilde{\pi}^{\theta^i}$ and must be computed at the server.

## 3.2 BCFL COMMUNICATION ROUNDS AS RECURSIVE BAYESIAN UPDATES

FL typically involves multiple communication rounds, where the server shares its model parameters to the clients such that local updates can be performed. This process is performed repeatedly such that the model learning is improved over iterations, which we denote by index $t$. Similarly, BCFL can be implemented over multiple communication rounds and this section formulates the recursive update of the main equations described in Section 3.1, as well as the associated challenges.

Following a Bayesian framework, we focus in computing the posterior distribution $p(\Omega|\mathcal{D}_{1:t})$ of the set of parameters given all available data up to iteration $t$, that is $\mathcal{D}_{1:t} \triangleq \{\mathcal{D}_{1:t}^1, \dots, \mathcal{D}_{1:t}^C\}$ with $\mathcal{D}_{1:t}^j \triangleq \{\mathcal{D}_1^j, \dots, \mathcal{D}_t^j\}$ for the $j \in \{1, \dots, C\}$ client. If a finite number of iterations $T$ are performed, then the iteration index is an integer $t \in [0, T]$ such that $t = 0$ denotes the initialization step of BCFL. Notice that $T$ can be arbitrarily large, or even infinite-horizon when $T \to \infty$. This definition of the datasets $\mathcal{D}_t^j$ encompasses different situations. For instance, $\mathcal{D}_t^j$ could be randomly drawn from the same random local distribution or be the same dataset at every iteration.

To recursively compute the posterior at iteration $t$, the latest available posterior at the server $p(\Omega|\mathcal{D}_{1:t-1})$ becomes the a priori distribution $p(\Omega)$ for the Bayesian update in equation 1, that is $p(\Omega|\mathcal{D}_{1:t}) \propto p(\mathcal{D}_t|\Omega)p(\Omega|\mathcal{D}_{1:t-1})$. For the sake of simplicity, we define $p_t(\Omega) \triangleq p(\Omega|\mathcal{D}_{1:t})$ to obtain more compact recursive expressions. We are interested in a mixture posterior representation as in equation 2, which in the recursive case results in

$$p_t(\Omega) = \sum_{\theta_{1:t}\in\Theta_{1:t}} \pi^{\theta_{1:t}} p_t^{\theta_{1:t}}(\Omega) \,, \tag{10}$$

where $\Theta_{1:t} = \Theta_1 \times \cdots \times \Theta_t$ is the set of all client/cluster associations until iteration $t$ such that $\sum_{\theta_{1:t} \in \Theta_{1:t}} = \sum_{\theta_1 \in \Theta_1} \sum_{\theta_2 \in \Theta_2} \cdots \sum_{\theta_t \in \Theta_t}$ and analogously as in equation 2 we have that $\pi^{\theta_{1:t}} = \mathbb{P}[\theta_{1:t}|\mathcal{D}_{1:t}]$ and $p_t^{\theta_{1:t}}(\Omega) = p(\Omega|\mathcal{D}_{1:t}, \theta_{1:t})$ since equation 10 can be interpreted as $p_t(\Omega) = \sum_{\theta_{1:t} \in \Theta_{1:t}} p(\Omega|\mathcal{D}_{1:t}, \theta_{1:t}) \mathbb{P}[\theta_{1:t}|\mathcal{D}_{1:t}]$. For a more compact notation, let us use $h \triangleq \theta_{1:t-1}$ and $\mathcal{H}_{t-1} \triangleq \Theta_{1:t-1}$ to denote a particular choice of past associations and the set of all possible past associations, respectively, such that $\Theta_{1:t} = \mathcal{H}_{t-1} \times \Theta_t$.

The mixture posterior in equation 10 results from considering that the prior distribution at $t$ is itself a mixture distribution containing all possible associations up to $t - 1$, we can then write

$$p_t(\Omega) \propto p(\mathcal{D}_t|\Omega)p_{t-1}(\Omega) = \left( \sum_{\theta_t \in \Theta_t} p(\mathcal{D}_t, \theta_t|\Omega) \right) \left( \sum_{h \in \mathcal{H}_{t-1}} \pi^h p_{t-1}^h(\Omega) \right)$$

$$= \sum_{h \in \mathcal{H}_{t-1}} \sum_{\theta_t \in \Theta_t} \pi^h p(\mathcal{D}_t, \theta_t|\Omega) p_{t-1}^h(\Omega) \tag{11}$$

and equation 10 appears considering the Bayes update $p_t^{\theta_{1:t}}(\Omega) = p(\mathcal{D}_t, \theta_t|\Omega)p_{t-1}^h(\Omega)/\pi^{\theta_t|h}$ under association hypothesis $\theta_{1:t}$ and corresponding unnormalized weight $\tilde{\pi}^{\theta_{1:t}} = \pi^h \tilde{\pi}^{\theta_t|h}$, where the normalizing constant is $\tilde{\pi}^{\theta_t|h} = \int p(\mathcal{D}_t, \theta_t|\Omega)p_{t-1}^h(\Omega)d\Omega$. The normalized weights are then $\pi^{\theta_{1:t}} = \tilde{\pi}^{\theta_{1:t}}/\sum_{h \in \mathcal{H}_{t-1}} \sum_{\theta_t \in \Theta_t} \tilde{\pi}^{\theta_{1:t}}$. Notice that, regardless of the cluttered notation due to the past and present hypotheses, the posterior update at $t$ can be computed as detailed in Section 3.1 where it was shown that this can be achieved using only local computations, which are aggregated at the server. Algorithm 1 presents the pseudo-code for the conceptual BCFL approach.

### 3.3 THE EXPLOSION OF ASSOCIATION HYPOTHESES

A major issue regarding the Bayesian framework presented above is the quick growth of association hypothesis over communication rounds $t$. At a given iteration, the number of possible associations for $K$ clusters and $C$ clients is given by $N(K, C) = \prod_{i=1}^{K} \sum_{c=0}^{C} \mathcal{C}(C, c)$, where $\mathcal{C}(C, c) = \frac{C!}{c!(C-c)!}$ represents the number of $c$ element combinations out of $C$ elements. Furthermore, the number of hypotheses in the posterior distribution increases very rapidly due to the recursive training. That is, at a given iteration $t$, the number of possible clusters corresponds to the modes of the shared prior distribution, $N_t(K_{t-1}, C) = |\mathcal{H}_{t-1}| = \prod_{\tau=1}^{t} N_\tau(K_{\tau-1}, C)$, causing $N(K_t, C)$ to explode. Therefore, due to this *curse of dimensionality*, we observe that evaluation of the exact posterior is intractable and that approximations are necessary in order to design efficient algorithms that approximate the conceptual solution provided by equation 10. The proposal of such algorithms, based on three different approximations, is the purpose of Section 4. In general, we aim at finding a subset of data associations $\hat{\Theta}_t \subset \Theta_t$, such that $|\hat{\Theta}_t| \ll |\Theta_t|$ while a cost is minimized to ensure relevant associations are kept, thus formulating the choice of associations as an optimization problem.

## 4 APPROXIMATE BCFL: SIMPLIFIED DATA ASSOCIATION STRATEGIES

To address the intractability of the number of association hypotheses, this section presents three different strategies to select subsets of associations hypotheses, as well as their corresponding practical algorithms. In particular, this section discusses the metric used to quantify the cost of associating a client to a cluster, which is then used to make decisions on the desired association subsets $\hat{\Theta}_t$.

A key assumption that is enforced to limit the number of associations is that at any iteration $t$ a client can only be associated with one cluster for a given hypothesis. Notice however that since multiple association hypotheses are considered, every client has a chance to being associated with different clusters. This assumption implies that there are no duplications in the different client partitions, thus, dramatically reducing the number of possible associations. Additionally, we assume that every selected client must be associated with a cluster such that all data is used.

### 4.1 THE COST OF DATA ASSOCIATION

Before diving into different association selection methods it is paramount to determine a metric, or cost, for a particular association. In this paper, we adopt an assignment problem formulation (Alfaro et al., 2022). Let $L \in \mathbb{R}^{C \times K}$ be a cost matrix whose entries $L^{j,i}$ represent the cost of assigning the

$j$-th client to the $i$-th cluster, and $A$ be a binary assignment matrix with entries $A^{j,i} \in \{0, 1\}$. The total cost of a given association $A$ can be written as $\text{Tr}(A^\top L)$. The assumption that a client's data can only be associated with one cluster can be imposed with the constraint $\sum_{i=1}^{K} A^{j,i} = 1, \forall j$ over the association matrix $A$. In general, we would like to select associations with the smallest possible cost. Obtaining the best association according to the cost $\text{Tr}(A^\top L) = \sum_{i=1}^{K} \sum_{j=1}^{C} A^{i,j} L^{j,i}$ can be formalized as a constrained optimization problem of the form

$$A^\star = \arg\min_{A} \sum_{i=1}^{K} \sum_{j=1}^{C} A^{i,j} L^{j,i}, \quad \text{s.t.} \quad A^{i,j} \in \{0,1\}, \text{ and } \sum_{i=1}^{K} A^{j,i} = 1, \forall j , \quad (12)$$

for which efficient algorithms exist such as the Hungarian or the Auction algorithms (Fredman & Tarjan, 1987), as well as the Jonker-Volgenant-Castanon algorithm (Jonker & Volgenant, 1988; Drummond et al., 1990). Remarkably, a naive solution to the optimization has factorial complexity, while those algorithms can solve the assignment problem in polynomial time. The optimization of equation 12 results in the optimal association matrix $A^\star$, which is equivalent to the optimal association variable $\tilde{\pi}^{\theta_t^\star | h}$ such that

$$\tilde{\pi}^{\theta_t^\star | h} \geq \tilde{\pi}^{\theta_t | h} , \forall \theta_t \in \Theta_t \quad (13)$$

given past associations $h$. A related problem is to find the $M$ best assigments to the problem in equation 12, where notice that the previous case corresponds to $M = 1$. In that case, one could leverage Murty's algorithm (Miller et al., 1997) to rank the solutions

$$\tilde{\pi}^{\theta_{t,(1)}^\star | h} \geq \tilde{\pi}^{\theta_{t,(2)}^\star | h} \geq \cdots \geq \tilde{\pi}^{\theta_t^\star | h} \geq \tilde{\pi}^{\theta_{t,(M)} | h} , \forall \theta \in \Theta_t \backslash \{\theta_{t,(m)}^\star\}_{m=1}^{M} \quad (14)$$

Up to this point, the cost matrix $L$ has not been specifically defined. To do so, we first express the problem in equation 12 in terms of the association variable $\theta$, such that a reasonable choice for the cost function is to minimize the negative log-weights given $h$, which we denote by $\ell^{\theta_t | h} = -\log(\tilde{\pi}^{\theta_t | h})$. Recall that the weights are related to the association posterior given data and past associations, and thus minimizing $\ell^{\theta_t | h}$ corresponds to maximizing that posterior. In addition, we note that optimizing

$$\theta_t^\star | h = \arg\min_{\theta_t \in \Theta_t} \ell^{\theta_t | h} = \arg\min_{\theta_t \in \Theta_t} \sum_{i=1}^{K} \sum_{j \in \theta^i} -\log(\tilde{\pi}_j^{\theta_t^i | h}) \quad (15)$$

is equivalent to the assignment problem in equation 12 where the elements of the cost matrix $L$ are given by $\ell_j^{\theta_t^i | h} = -\log(\tilde{\pi}_j^{\theta_t^i | h})$ such that the assignments $A$ correspond to a unique $\theta_t \in \Theta_t$. As a consequence, we can use the aforementioned assigment algorithms to find sets of client-cluster associations, which is leveraged in the practical BCFL methods proposed next.

## 4.2 Greedy Association: BCFL-G

The most simplistic association is to follow a greedy approach where at each iteration $t$ only the best association is kept. Denoting as $\theta^\star$ the association leading to the optimal assignment, see equation 13, and denoting the sequence of optimal associations by $\theta_{1:t}^\star \triangleq \{\theta_1^\star, \theta_2^\star | \theta_1^\star, \ldots, \theta_t^\star | \theta_{1:t-1}^\star\}$, the posterior in equation 10 can be approximated as

$$p_t^{\text{G}}(\Omega) = p_t^{\theta_{1:t}^\star}(\Omega) = \prod_{i=1}^{K} p_t^{\theta_{1:t}^{\star,i}}(\omega^i) \quad (16)$$

where we have $\pi^{\theta_{1:t}} = \pi^{\theta_{1:t}^\star} = 1$, since only the best association is kept at each time $t$. The posterior in equation 16 can be recursively obtained from the posterior of the previous time step, that is, $p_t^{\theta_{1:t}^{\star,i}}(\omega^i) \propto p(\mathcal{D}_t^{\theta_t^\star,i} | \omega_t^i) p_{t-1}^{\theta_{1:t-1}^{\star,i}}(\omega^i)$ as discussed in Section 3.2. Note that unnormalized local updates of the posterior can be computed locally at the clients while aggregation and normalization need to be performed in the server. Algorithm 2 presents the pseudo-code for BCFL-G.

The greedy approach has several benefits, mostly related to its reduced complexity which makes it computationally cheap and relatively simple to implement. The downside is that there is no guarantee that the selected trajectory of hypotheses – which are the best for a given iteration conditional on past associations – is optimal in a broader sense of $\tilde{\pi}^{\theta_{1:t}^\star} \geq \tilde{\pi}^{\theta_{1:t}} , \forall \theta_{1:t} \in \Theta_{1:t}$. Therefore, this points out that keeping a specific association only might not be sufficient to represent the uncertainty of the random variable $\theta_{1:t}$, which motivates the next two practical strategies.

The resulting BCFL-G, for *greedy*, association strategy is somewhat correlated with the method proposed in Ghosh et al. (2020), despite their deterministic perspective, in which the best clusters

are selected at every time step by maximizing the data-fit at each client. However, regardless the existing similarities, we would like to highlight that the Bayesian framework proposed in this paper is more general, which enables many other association strategies as will be discussed next.

### 4.3 CONSENSUS ASSOCIATION: BCFL-C

Alternatively to keeping the best instantaneous association, while still keeping the number of associations small, a further enhancement is to merge the $M$ best associations and keep that single association instead. We refer to this the *consensus* approach, or BCFL-C for short. This additional aggregation of hypotheses improves the uncertainty characterization of $\theta_{1:t}$, which in turn results in better performance results as is discussed in the results section. Noticeably, the computational complexity of BCFL-C is comparable to that of BCFL-G, while it has the same lack of optimality guarantees in a broader sense due to the assumptions of keeping on hypothesis over iterations. Following similar definitions as in Section 4.2, we denote the resulting posterior approximation as

$$p_t^{\mathrm{C}}(\Omega) = \prod_{i=1}^{K} \mathsf{MERGE} \left( \sum_{m=1}^{M} \pi^{\theta_{t,(m)}^{\star,i}|h} p_{t|t-1}^{\theta_{t,(m)}^{\star,i}|h}(\omega^i) \right) \tag{17}$$

where $p_{t|t-1}^{\theta_{t,(m)}^{\star,i}|h}(\omega_t^i)$ is the posterior distribution of cluster $i$ given the $m$-th best instantaneous hypothesis $\theta_{t,(m)}^{\star,i}$ and past associations $h$. The $\mathsf{MERGE}(\cdot)$ operator is a function that fuses the $M$ into a single density, which can be accomplished by moment matching or other techniques (Bishop & Nasrabadi, 2006). For Gaussian densities, this can be easily obtained (Li et al., 2019; Luengo et al., 2018), as shown in Appendix C. Algorithm 3 presents the pseudo-code for BCFL-C.

### 4.4 MULTIPLE ASSOCIATION HYPOTHESIS: BCFL-MH

A relaxation of the approximations performed by BCFL-G and BCFL-C, where a single hypothesis is propagated over iterations, is to keep track of several trajectories of possible association hypotheses. The general posterior in equation 10 then results in the *multi-hypothesis* approach BCFL-MH:

$$p_t^{\mathrm{MH}}(\Omega) = \sum_{\theta_{1:t} \in \hat{\Theta}_{1:t}} \pi^{\theta_{1:t}} p_t^{\theta_{1:t}}(\Omega) , \tag{18}$$

which in essence implies finding the subset $\hat{\Theta}_{1:t} \subset \Theta_{1:t}$ of $M_{\max} = |\hat{\Theta}_{1:t}|$ highly-promising hypotheses that the method would update. The identification of this subset and its recursive update can be performed by pruning associations with small weights below a predefined threshold, then use the Murty's algorithm or similar to rank the best $M_{\max}$ associations. BCFL-MH is arguably more complex to implement due to the need for keeping track of multiple association hypotheses. However, we will see that its performance is typically superior since for large $M_{\max}$ values the uncertainty of associations hypotheses can be accurately characterized. The BCFL-MH distribution in equation 18 is then parameterized by weights and densities for each of the $M_{\max}$ trajectories selected at round $t$, $\{\pi^{\theta_{1:t}}, \{p_t^{\theta_{1:t}^i}(\omega^i)\}_{i=1}^{K}\}_{\theta_{1:t} \in \hat{\Theta}_{1:t}}$. For a given hypothesis, the weight and densities are computed as described in Sections 3.1 and 3.2. Algorithm 3 presents the pseudo-code for BCFL-MH.

## 5 EXPERIMENTS

To validate the proposed BCFL methods under both feature- and label-skew situations, we generate four non-IID scenarios using four well-known datasets: Digits-Five, AmazonReview, Fashion-MNIST and CIFAR-10 (see Appendix D.1). Digits-Five and AmazonReview datasets contain data coming from different categories, making them suitable to test feature-skewed situations. To generate the feature-skewed scenario we split data with different characteristics among multiple clients. We create two scenarios using Digits-Five and AmazonReview datasets since data from these datasets are already categorized into different groups. For the Digits-Five scenario, we split the data among $C = 10$ clients, 2 per sub-dataset, leading to 5 disjoint client groups. For the AmazonReview scenario, we split the data among $C = 8$ clients, 2 per merchandise category. As for label-skewed data, we generate two scenarios using Fashion-MNIST and CIFAR-10, generating label-skewed groups using a two-stage Dirichlet-based sampling approach (Ma et al., 2022), process that is controlled by the concentration parameter $\alpha$ which is set to $0.1$ in our experiments. More details regarding this sampling process can be found in Appendix D.2.

**Baseline models and system settings.** We selected two competing methods for benchmark purposes. They are the well-known FedAvg (McMahan et al., 2017), which is a single-model-based FL strategy, and WeCFL which is the state-of-the-art clustered FL method (Ma et al., 2022). In this work, we consider the training of a neural network (NN) model using FL, in which case the local posteriors in equation 8 are obtained using Laplace approximation as in Liu et al. (2021) and the weights in equation 9 are related to the corresponding training loss. Details on the system settings can be found in the Appendix D.2. **Model warm-up.** In the experiments we evaluate the impact of model warm-up, whose purpose is to improve the initialization of the local models, potentially improving the overall FL solution. In this work, warm-up is implemented in two steps: first using a Euclidean-distance metric to cluster the parameters of local models and then aggregating them into a merged model, which is then shared among those in the same cluster. Experiments reveal that, while warm-up can be beneficial in terms of convergence time and accuracy, the BCFL schemes exhibit competitive performances without thus making warm-up not strictly necessary. **Evaluation metrics:** We evaluate the performance using both micro accuracy (acc %) and macro F1-score (F1) on the client-wise test datasets due to high non-IID degrees. Micro average is performed for accuracy to balance the different number of samples in the different clients.

Table 1: Performance comparison on cluster-wise non-IID. For Digits-Five and AmazonReview feature-skewed data scenarios, Feature $(K, C/K)$ indicates $K$ data groups with $C/K$ clients per group. For Fashion-MNIST and CIFAR-10 label-skewed data scenarios, Label $(K, C/K, \alpha)$ indicates also the value of the Dirichlet concentration parameter $\alpha$.

| Datasets | Digits-Five | | AmazonReview | | Fashion-MNIST | | CIFAR-10 | |
|---|---|---|---|---|---|---|---|---|
| non-IID setting | Feature $(5,2)$ | | Feature $(4,2)$ | | Label $(4,10,0.1)$ | | Label $(4,10,0.1)$ | |
| Methods | Acc | F1 | Acc | F1 | Acc | F1 | Acc | F1 |
| FedAvg | 93.18 | 92.95 | 87.71 | 87.70 | 85.00 | 53.14 | 39.89 | 21.19 |
| WeCFL | 93.81 | 93.58 | 85.57 | 85.51 | 96.74 | 92.0 | 72.91 | 51.69 |
| BCFL-G | 94.35 | 94.16 | 87.53 | 87.5 | 96.81 | 93.51 | 64.73 | 44.22 |
| BCFL-G-W | 94.48 | 94.29 | 87.58 | 87.57 | 96.69 | 92.21 | 72.95 | 52.01 |
| BCFL-C-3 | 94.04 | 93.84 | 87.95 | 87.93 | 96.84 | 90.16 | 72.12 | 50.97 |
| BCFL-C-3-W | 94.35 | 94.17 | 88.22 | 88.20 | 96.88 | 90.06 | 73.18 | 52.61 |
| BCFL-C-6 | 94.14 | 93.93 | 88.11 | 88.09 | 96.58 | 86.55 | 68.73 | 47.40 |
| BCFL-C-6-W | 94.42 | 94.24 | 87.96 | 87.95 | 96.71 | 87.31 | 72.22 | 50.02 |
| BCFL-MH-3 | 95.39 | 95.22 | **88.74** | **88.74** | 97.13 | **93.70** | 74.35 | 56.24 |
| BCFL-MH-3-W | 95.83 | 95.70 | 88.38 | 88.38 | **97.40** | 93.42 | 76.19 | 58.42 |
| BCFL-MH-6 | 96.02 | 95.88 | 88.16 | 88.15 | 97.27 | 92.56 | 75.26 | 53.45 |
| BCFL-MH-6-W | **96.22** | **96.08** | 88.43 | 88.43 | 97.33 | 90.62 | **77.56** | **59.18** |

## 5.1 Experimental Results

**Comparison study.** Table 1 provides insights into the performance of different methods on various datasets. We refer to the BCFL methods by their acronyms and append an '-W' to denote the use of warm-up. For BCFL-C and BCFL-MH, a number is also included to denote the number of associations considered, $M$ and $M_{max}$, respectively. We can notice the superior performance of

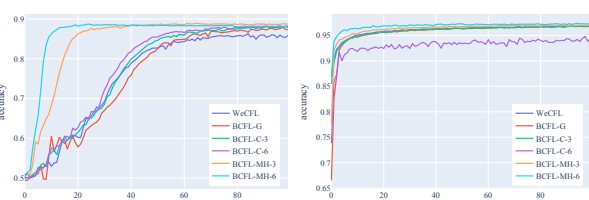

Figure 2: Accuracies for AmazonReview (left panel) and Fashion-MNIST (right panel) datasets.

the proposed BCFL variants. In fact, all the best results were obtained by the BCFL-MH class of methods in all datasets. We highlight, however, that the BCFL-MH variants are inherently more costly since they explore multiple association hypotheses throughout iterations. The less expensive BCFL-G and BCFL-C present results that are close to the results obtained BCFL-MH in most datasets and slightly superior to results obtained with WeCFL. FedAvg, which serves as a baseline, is surpassed by WeCFL in terms of performance, except for AmazonReview dataset. This suggests that AmazonReview dataset may not exhibit strong non-IID characteristics, given that a centralized

model like FedAvg can outperform other methods, including BCFL-G. Methods that incorporate warm-up tend to show slightly improved results compared to those without warm-up. Nevertheless, this indication of improvement should be taken lightly as both exhibit comparable results.

**Convergence analysis.** Figure 2 shows the convergence curves of several clustered FL methods including WeCFL, and multiple BCFL variants (G, C-3, C-6, MH-3, and MH-6, all without warm-up) for the AmazonReview and Fashion-MNIST datasets. It indicates that BCFL-MH-6 exhibits the fastest convergence rate among all methods. Indeed, according to our practical experience, BCFL-MH converges faster than the other methods in general. We evaluate the effect of using a warm-up stage for the BCFL methods. The convergence curves depicted in Figure 3 indicate that warm-up can slightly accelerate convergence. Similar figures for the other datasets are provided in Appendix D.3.

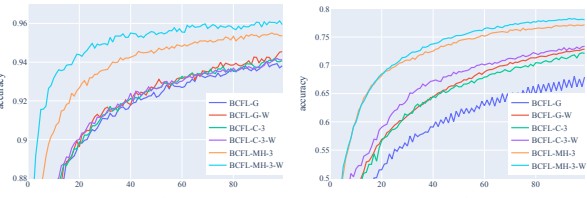

Figure 3: Warm-up comparison of accuracy fir Digits-Five (left panel) and CIFAR-10 (right panel)

**Clustering analysis.** To better understand the dynamics of information-sharing among clients, we visualize the clients' cluster graph across all rounds. Figure 4 focuses on the Digits-Five dataset, similar analysis for the other experiments can be found in Appendix D. In the figure, the thickness of the edges between clients represents the degree of information sharing between them during the training process (i.e. their accumulated probability of associating to the same cluster). The graph shows that while WeCFL indeed converges to groups, BCFL-G exhibits more client connections and potential for information exchange. With even more client connections, BCFL-MH formed connections among almost all clients. Nevertheless, we can notice that stronger connections were formed between client groups $\{0,1\}$, $\{2,3\}$, $\{4,5\}$ and $\{6,7,8,9\}$, correctly clustering clients observing similar features. There is a considerable connection between groups $\{2,3\}$ and $\{4,5\}$ as well, which could be considered a cluster itself.

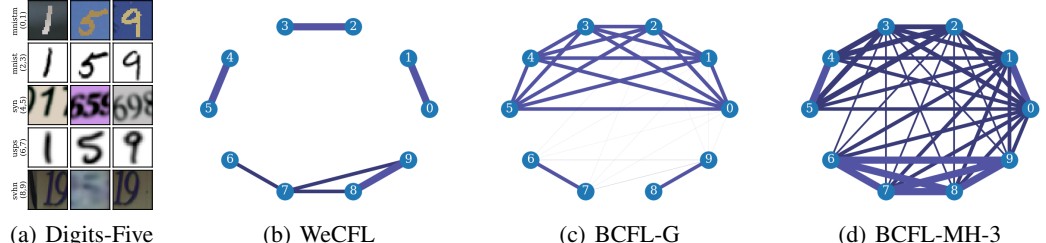

| (a) Digits-Five | (b) WeCFL | (c) BCFL-G | (d) BCFL-MH-3 |

Figure 4: Client clustering during training for Digits-Five dataset and selected CFL methods. $K = 5$ clusters are pairwise split into $C = 10$ clients, as denoted on the left labeling in (a).

## 6    CONCLUSION

This work presents a unifying framework for clustered Federated Learning (FL) that employs probabilistic data association to infer the relation between clients and clusters. The framework shows the conceptually optimal solution to the problem and highlights the need for approximations to make its implementation feasible. It paves the way to new research solutions to address the long-standing challenge of handling non-IID data in FL systems. In particular, three different approximations to reduce the number of association hypotheses are proposed, with different complexity requirements, exhibiting competitive results that are superior to state-of-the-art methods. Additionally, the probabilistic approach of the framework provides uncertainty measures that are seen to enable cross-client/cluster pollination that enhanced its performance. Future work includes extending the BCFL approach to estimate the number of clusters, which has the added challenge that the parameter space has an unknown dimension.

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

# APPENDICES

## A    NOTATION

Table 2: Table of relevant notation conventions.

| Notation | Definition |
|---|---|
| $K, i$ | Number of clustered models in server and cluster model index. |
| $C, j$ | Number of local clients and client index. |
| $\omega^i, \Omega$ | $i - th$ cluster model parameters and combined parameters for the $K$ clusters. |
| $t$ | FL communication round index |
| $\mathcal{D}^j, \mathcal{D}$ | $j$-th client local dataset and combined dataset for all clients. |
| $\theta^i, \theta, \Theta$ | Client association indices for cluster $i$, combined association hypotheses for all clusters, and set of all associations. |
| $\pi^\theta, \tilde{\pi}^\theta$ | Normalized and unnormalized posterior weights for association $\theta$. |
| $\pi_j^{\theta^i}, \tilde{\pi}_j^{\theta^i}$ | Normalized and unnormalized local likelihood of client $j$ given cluster $i$. |
| $\pi_{j,t}^{\theta^i}, \tilde{\pi}_{j,t}^{\theta^i}$ | Normalized and unnormalized local likelihood of client $j$ given cluster $i$ in communication round $t$. |
| $p^\theta(\Omega), p(\Omega)$ | Posterior density given the association hypothesis $\theta$, and full posterior. |
| $p^{\theta^i}(\omega^i), \pi^{\theta^i}$ | Posterior density of cluster $i$ given the association hypothesis $\theta$ and its weight. |
| $\mathcal{D}_{1:t}^j, \mathcal{D}_{1:t}$ | Client $j$ dataset from iterations 1 to $t$, and combined dataset from all clients. |
| $\Theta_{1:t}, \theta_{1:t}$ | Set of all possible hypotheses from $1:t$ and a particular association. |
| $\pi^{\theta_{1:t}}, p_t^{\theta_{1:t}}(\Omega)$ | Weights and posterior given association $\theta_{1:t}$. |
| $\mathcal{H}_{t-1}, h$ | Set of past associations and a particular one. |
| $\pi^h$ | Accumulated weights from past $h$ associations. |
| $N(K, C)$ | Number of total associations given $K$ clusters and $C$ clients. |
| $A, L$ | Assignment and cost association matrices. |
| $\theta_t^\star \mid h$ | Optimal association at $t$ given past associations $h$. |
| $\theta_{t,(m)}^\star$ | $m$-th best association at $t$ given past associations $h$. |
| $\ell^\theta$ | Negative posterior log-weight given $\theta$. |
| $p_t(\Omega)$ | Full posterior at $t$. |
| $p_t^{\mathrm{C}}(\Omega)$ | BCFL-C posterior approximation at $t$. |
| $p_t^{\mathrm{G}}(\Omega)$ | BCFL-G posterior approximation at $t$. |
| $p_t^{\mathrm{MH}}(\Omega)$ | BCFL-MH posterior approximation at $t$. |

## B    ALGORITHMS

In this appendix, we bring the pseudo-code for the conceptual BCFL algorithm 1, which retains all possible associations at each iteration, and the approximations with pseudo-codes given in Algorithm 2 for BCFL-G and Algorithm 3 for both BCFL-C and BCFL-MH.

Note that in the conceptual algorithm 1, there is no need for identifying specific subsets of associations, instead, all associations are retained. This approach allows for the simultaneous update of association weights and the local posterior. Conversely, the approximated algorithm requires a pre-selection of associations to be preserved, necessitating a decision prior to local model training. Consequently, the process involves two communication rounds: the first to upload association weights and confirm the decision, followed by a second where the decision is downloaded locally to guide training, thereby reducing superfluous computational expenses.

For algorithm BCFL-G 2, $M_{\mathrm{max}} = 1$, the generated hypothesis will be singular, significantly simplifying computations. Moreover, efficiency gains can be achieved by localizing decision-making, which eliminates the need for one round of communication since the optimal association is selected through a greedy approach known locally, removing the necessity to transmit data back to the server. This process aligns with the IFCA method introduced by Ghosh et al. (2020), as detailed in Algorithm 2.

Regarding the other two approximations BCFL-C and BCFL-MH, see Algorithm 3, the complexity is mitigated by keeping only the most promising $M_{\mathrm{max}}$ hypotheses under the assumption that, in each

hypothesis, one client can only be associated to one cluster. If all updated hypotheses are merged into a single one for the next iteration, we call it BCFL-C, whereas if we keep $M_{\max}$ for the next iteration, it is referred to as BCFL-MH, as described in Algorithm 3.

In the case of the BCFL-C, where $M_{\max}$ is not equal to 1, the server's role is to merge all $M_{\max}$ hypotheses. Detailed information on how this merging is executed can be found in the Appendix C.

As for the BCFL-MH algorithm, where $M_{\max}$ are not equal to 1, pruning is performed maintaining $M_{\max}$ hypotheses with largest weights.

---

**Algorithm 1** Conceptual BCFL algorithm

---

**Input:** $K$
**Initialization:** $\mathcal{H}_0$, $\{\{\pi_0^{\theta^i}, p_0^{\theta^i}(\omega^i)\}_{i=1}^K\}_{\theta \in \mathcal{H}_0}$;
**for** $t = 1, \cdots, T$ **do**
    **Server do**
        Broadcast $\{\{p_{t-1}^{\theta^i|h}(\omega^i)\}_{i=1}^K\}_{\theta \in \mathcal{H}_{t-1}}$ to participating clients
    **end Server**
    **for** Client $j \in \{1, \ldots, C\}$ in parallel **do**
        **for** every hypothesis $h \in \mathcal{H}_{t-1}$ in parallel **do**
            Compute association weight $\{\tilde{\pi}_{j,t}^{\theta^i|h}\}_{i=1}^K$ as in equation 9
            Compute posterior update $\{p_t^{\theta^i|h}(\omega^i|\mathcal{D}^j)\}_{i=1}^K$ for each cluster $i$
            Transmit $\{\tilde{\pi}_{j,t}^{\theta^i|h}, p_t^{\theta^i|h}(\omega^i|\mathcal{D}^j)\}_{i=1}^K$ to the server
        **end for**
    **end for**
    **Server do**
        Update hypothesis set $\mathcal{H}_t = \mathcal{H}_{t-1} \times \Theta_t$
        Compute $\{p_t^{\theta^i|h}(\omega^i)\}_{i=1}^K$ as in equation 8
        Compute posterior $p_t^{\theta_{1:t}}(\Omega) \propto \prod_{i=1}^K p_t^{\theta^i|h}(\omega^i)$, for all $\theta_t \in \Theta_t$ and $h \in \mathcal{H}_{t-1}$
        Compute $\{\pi_t^{\theta^i|h}\}_{i=1}^K$ as in equation 9 and obtain $\pi^{\theta_{1:t}} \propto \pi^h \tilde{\pi}^{\theta_t|h}$
        Compute full posterior $p_t(\Omega)$ as in equation 11
    **end Server**
**end for**
**return:** $p_t(\Omega)$

---

**Algorithm 2** BCFL-G algorithm

---

**Input:** $K$
**Initialization:** $\{p_0^i(\omega^i)\}_{i=1}^K$;
**for** $t = 1, \cdots, T$ **do**
    **Server do**
        broadcast $\{p_{t-1}^i(\omega^i)\}_{i=1}^K\}$ to selected clients
    **end Server**
    **for** Client $j \in \{1, \ldots, C\}$ in parallel **do**
        Compute association weight $p_t(\mathcal{D}^j|\omega_1^i)$
        Compare weights across clusters and get the best assignment
        Compute the local posterior $p_t^*(\omega^i|\mathcal{D}^j)$ and transmits it to the server
    **end for**
    **Server do**
        Compute the cluster posteriors $\{p_t^{\theta_{1:t}^{\star,i}}(\omega^i)\}_{i=1}^K$
        Compute the posterior $p_t^G(\Omega)$ as in equation 16
    **end Server**
**end for**
**Return:** $p_t^G(\Omega)$

---

---

**Algorithm 3** Approximate BCFL-C and BCFL-MH algorithms

---

**Input:** $K$, $M_{\max}$
**Initialization:** $\mathcal{H}_0$, $\{\{\pi_0^{\theta^i}, p_0^{\theta^i}(\omega^i)\}_{i=1}^K\}_{\theta \in \mathcal{H}_0}$;
**for** $t = 1, \cdots, T$ **do**
   **Server do**
      Broadcast $\{\{p_t^{\theta^i|h}(\omega^i)\}_{i=1}^K\}_{\theta \in \mathcal{H}_{t-1}}$ to participating clients
   **end Server**
   **for** Client $j \in \{1, \ldots, C\}$ in parallel **do**
     **for** every hypothesis $h \in \mathcal{H}_{t-1}$ in parallel **do**
        Compute association weight $\{\tilde{\pi}_{j,t}^{\theta^i|h}\}_{i=1}^K$ as in equation 9
        Transmit weights to the server
     **end for**
   **end for**
   **Server do**
      Construct cost matrix $L$ as in section 4.1
      Compose $\Theta_t$ by keeping the $M_{\max}$ best associations
      $\mathcal{H}_t = \Theta_t \times \mathcal{H}_{t-1}$,
   **end Server**
   **for** $\theta \in \mathcal{H}_t$ in parallel **do**
     **for** $i$ in $K$ **do**
       **Server do**
          Broadcast association decision $\theta^i$ to associated clients
       **end Server**
       **for** Client $j \in \theta^i$ in parallel **do**
          Compute local posterior $p_t^{\theta^i|h}(\omega^i|\mathcal{D}^j)$ and transmit to the server
       **end for**
       **Server do**
          Compute $p_t^{\theta^i|h}(\omega^i)$, $\pi_t^{\theta^i|h}$ using equations 8 and 9
       **end Server**
     **end for**
   **end for**
   **Server do**
     **if** BCFL-C **then**
       Merging $M_{\max}$ best hypothesis into one, $|\mathcal{H}_t| = 1$
     **end if**
     **if** BCFL-MH **then**
       Pruning $M_{\max}$ best hypotheses with largest posterior weights, $|\mathcal{H}_t| = M_{\max}$
     **end if**
   **end Server**
**end for**
**Return:** $p_t^C(\Omega)$ or $p_t^{\mathrm{MH}}(\Omega)$

---

**Communication cost.** Federated Learning incurs notable communication costs due to the regular transmission of model updates between numerous decentralized clients and a central server, with costs influenced by factors such as the number of clients, model size, update frequency, data distribution, channel quality, and so on. In our communication analysis within this study, we focus exclusively on quantifying the volume of parameters that must be transmitted during each communication round. As previously addressed, the association weights are transmitted initially, with the quantity of weights sent per round being $KCM_{\max}$. In the case of BCFL-G, this transmission round is omitted, whereas for BCFL-C, it entails $KC$. Regarding the transmission of model parameters, the requisite size is $mKM_{\max}$, where $m$ represents the size of an individual model. Notably, BCFL-MH incurs a significantly higher computation cost, amounting to $M_{\max}$ times that of other methods.

Additionally, the BCFL-MH algorithm incurs a higher computational cost during local training compared to alternative methods. Consequently, its practicality in distributed systems that prioritize efficiency is limited. However, as demonstrated by the results, BCFL-MH is capable of outperforming

its counterparts. Therefore, distributed systems where computational cost is not a primary concern could leverage BCFL-MH to enhance performance, which is a significant consideration. When efficiency is of greater importance, a greedy and consensus approach may be adopted to balance the trade-off between performance and computational expense.

## C  MERGE($\cdot$) OPERATOR: FUSION OF GAUSSIAN DENSITIES

As the MERGE operator used in BCFL-C algorithm we considered the arithmetic averaging (AA) aggregation approach discussed in Li et al. (2019). Thus, given a Gaussian mixture $p(x) = \sum_{j=1}^{M} \phi_j \mathcal{N}(x; m_j, S_j)$, with weights $\phi_j$, means $m_j$ and covariances $S_j$, the application of the MERGE operator returns a single Gaussian $\mathcal{N}(x; \widetilde{m}, \widetilde{S})$ whose parameters are given as:

$$\widetilde{m} = \sum_{j=1}^{M} \phi_j m_j, \tag{19}$$

$$\widetilde{S} = \sum_{j=1}^{M} \phi_j (S_j + m_j m_j^\top - \widetilde{m}\widetilde{m}^\top). \tag{20}$$

## D  ADDITIONAL EXPERIMENTAL DETAILS

### D.1  DATASETS

To construct our experimental scenarios we leverage four popular datasets from which we construct two feature- and two label-skewed experiments. The datasets are:

1. **Digits-Five** (Peng et al., 2019) consists of a collection of five popular digit datasets: MNIST (mt) (55000 samples), MNIST-M (mm) (55000 samples), Synthetic Digits (syn) (25000 samples), SVHN (sv)(73257 samples), and USPS (up) (7438 samples). Each digit dataset includes a different style of 0-9 digit images.

2. **AmazonReview** (Blitzer et al., 2007) AmazonReview is a dataset to tackle the task of identifying whether the sentiment of a product review is positive or negative. This dataset includes reviews from four different merchandise categories: Books (B) (2834 samples), DVDs (D) (1199 samples), Electronics (E) (1883 samples), and Kitchen and housewares (K) (1755 samples).

3. **Fashion-MNIST** (Xiao et al., 2017) consists of 70000 $28 \times 28$ grayscale images in 10 classes, with 60000 training images and 10000 test images.

4. **CIFAR-10** (Krizhevsky et al., 2009) provides 60000 $32 \times 32$ color images in 10 classes, with 6000 images per class.

### D.2  DETAILS OF EXPERIMENTAL SETTINGS

**Model related settings.**  The models trained with the different FL methods in the experiments are neural networks. Particularly, we use small Convolutional Neural Networks (CNNs) with 3 convolutional layers for the Digits-Five dataset; 3 layers fully-connected layers for the AmazonReview dataset; and 2 convolutional layers for both Fashion-MNIST and CIFAR-10 datasets. More details refer to Tables 3–6. The optimization of the CNN was done using SGD with a learning rate 0.005 and momentum 0.9, with a batch size of 32. For the training, we run 100 global communication rounds, and the local steps in each communication are 10. The warm-up steps is set to 2.

**FL settings**  To simulate label distribution skewness across clients, we use a method based on the use of a Dirichlet distribution for sampling labels, method that has been used in many recent FL studies (Li et al., 2021). Specifically, for a given client $j$, we define the probability of sampling data from the $k \in \{1, \ldots, l\}$ label as the vector $(p_{j,1}, \ldots, p_{j,l}) \sim \text{Dir}(\boldsymbol{\alpha})$, where $\text{Dir}(\cdot)$ denotes the Dirichlet distribution and $\boldsymbol{\alpha} = (\alpha_{j,1}, \ldots, \alpha_{j,l})$ is the concentration vector parameter ($\alpha_{j,k} > 0, \forall k$).

Table 3: Detailed information of CNN for Digits-Five.

| Layers | Details |
|---|---|
| Convolution | Conv2d(3, 64, kernel size = (5, 5), padding = 2)
BatchNorm2d(64)
ReLU()
MaxPool2d(3, 3) |
| Convolution | Conv2d(64, 64, kernel size = (5, 5), padding = 2)
BatchNorm2d(64)
ReLU()
MaxPool2d(3, 3) |
| Convolution | Conv2d(64, 128, kernel size = (5, 5), padding = 2)
BatchNorm2d(128)
ReLU() |
| Linear | Linear(6272, 3072)
BatchNorm1d(3072)
ReLU() |
| Linear | Linear(3072, 2048)
BatchNorm1d(2048)
ReLU() |
| Classifier | Linear(2048, 10) |
| Loss | CrossEntropy() |

Table 4: Detailed information of CNN for AmazonReview.

| Layers | Details |
|---|---|
| Linear | Linear(5000, 1000)
ReLU() |
| Linear | Linear(1000, 500)
ReLU() |
| Linear | Linear(500, 100) |
| Classifier | Linear(100, 2) |
| Loss | CrossEntropy() |

Table 5: Detailed information of CNN for Fashion-MNIST.

| Layers | Details |
|---|---|
| Convolution | Conv2d(1, 16, kernel size = (5, 5), padding = 2)
BatchNorm2d(16)
ReLU()
MaxPool2d(2, 2) |
| Convolution | Conv2d(16, 32, kernel size = (5, 5), padding = 2)
BatchNorm2d(16)
ReLU()
MaxPool2d(2, 2) |
| Classifier | Linear($7 \cdot 7 \cdot 32$, 10) |
| Loss | CrossEntropy() |

Table 6: Detailed information of CNN for CIFAR-10.

| Layers | Details |
|---|---|
| Convolution | Conv2d(3, 6, kernel size = (5, 5))
ReLU()
MaxPool2d(2, 2) |
| Convolution | Conv2d(6, 6, kernel size = (5, 5))
ReLU()
MaxPool2d(2, 2) |
| Linear | Linear(400,120)
ReLU() |
| Linear | Linear(120,84)
ReLU() |
| Classifier | Linear(84, 10) |
| Loss | CrossEntropy() |

Table 7: Performance comparison.

| Datasets | Fashion-MNIST | | AmazonReview | |
|---|---|---|---|---|
| Non-IID setting | Label $(0, 40, 0.1)$ | | Feature $(4, 10)$ | |
| Methods | Accuracy | F1 | Accuracy | F1 |
| FedAvg | 88.64 | 45.9 | 87.86 | 87.77 |
| WeCFL | 90.67 | 55.39 | 85.63 | 85.56 |
| BCFL-G | 92.79 | **69.13** | 88.01 | 87.95 |
| BCFL-G-W | 92.46 | 62.75 | 87.24 | 87.17 |
| BCFL-C-3 | 91.24 | 61.27 | **88.36** | **88.3** |
| BCFL-C-3-W | 92.39 | 60.87 | 87.38 | 87.34 |
| BCFL-C-6 | 92.17 | 60.76 | 88.24 | 88.18 |
| BCFL-C-6-W | 90.92 | 62.62 | 86.86 | 86.79 |
| BCFL-MH-3 | 93.41 | 67.48 | 87.21 | 87.14 |
| BCFL-MH-3-W | 93.49 | 63.42 | 87.12 | 87.06 |
| BCFL-MH-6 | **94.29** | 68.69 | 87.87 | 87.81 |
| BCFL-MH-6-W | 94.22 | 67.6 | 87.58 | 87.48 |

The advantage of this approach is that the imbalance level can be flexibly changed by adjusting the concentration parameter $\alpha_{j,k}$. If it is set to a smaller value, the partition is more unbalanced.

In our work, to generate our label-skewed dataset, we follow a two-step process as in Ma et al. (2022). First, we divide the dataset into four groups, with a common concentration parameter of $\alpha = 0.1$ for all the labels. This ensures that each group's distribution is different from the others. Then, we further divide the data into 10 clients per group, with a concentration parameter of $\alpha = 10$, to control that the clients from the same group have a similar distribution and thus could be clustered together.

### D.3 ADDITIONAL EXPERIMENTAL RESULTS.

We conducted additional experiments to those reported in the main body of the paper, the results of which are shown in Table 7 for Fashion-MNIST and AmazonReview. These experiments consider different label- and feature-skewed configurations to those in the experiments of the main body of the paper. For Fashion-MNIST we considered the Label$(0, 40, 0.1)$ configuration, indicating no groups, $C = 40$ clients, and $\alpha = 0.1$. For AmazonReview, we used a large number of clients (40) to showcase the results in larger datasets. In both cases, BCFL variants outperformed WeCFL and FedAvg.

We conducted additional experiments to study the sensitivity of the methods to misspecification of the number of cluster $K$. In practice, selecting the correct number of clusters might not be possible in certain applications, while in others we might have access to such information. The results of the sensitivity analysis to $K$ are shown in Tables 8 and 9, where the correct number of clusters is $K = 4$.

Table 8: Fashion-MNIST comparison of different $K$ with non-IID data setting Label$(4, 10, 0.1)$

| $K$ | 2 | | 3 | | 4 | | 5 | | 6 | |
|---|---|---|---|---|---|---|---|---|---|---|
| Methods | Acc | F1 | Acc | F1 | Acc | F1 | Acc | F1 | Acc | F1 |
| FedAvg | 85.00 | 53.14 | 85.00 | 53.14 | 85.00 | 53.14 | 85.00 | 53.14 | 85.00 | 53.14 |
| WeCFL | 91.79 | 75.99 | **96.74** | 88.92 | **96.74** | 92.00 | 96.65 | **92.21** | 96.73 | 91.98 |
| BCFL-G | 88.98 | 71.78 | 89.40 | 72.10 | 96.81 | **93.51** | **96.91** | 93.19 | 96.83 | 92.98 |
| BCFL-C-3 | 91.84 | 72.05 | 95.82 | 83.15 | **96.84** | 90.16 | 96.83 | **93.5** | 96.76 | 92.44 |
| BCFL-C-6 | 92.87 | 71.25 | 96.89 | 90.51 | 96.58 | 86.55 | **97.02** | **93.22** | 96.84 | 92.87 |
| BCFL-MH-3 | 94.39 | 74.60 | 97.23 | 92.53 | 97.13 | 93.70 | **97.32** | 93.18 | 97.35 | **94.02** |
| BCFL-MH-6 | 87.21 | 63.59 | 93.42 | 82.49 | **97.27** | **94.56** | 97.23 | 92.46 | **97.27** | 94.16 |

Table 9: AmazonReview comparison of different $K$ with non-IID data setting Feature$(4, 10)$

| $K$ | 2 | | 3 | | 4 | | 5 | | 6 | |
|---|---|---|---|---|---|---|---|---|---|---|
| Methods | Acc | F1 | Acc | F1 | Acc | F1 | Acc | F1 | Acc | F1 |
| FedAvg | 87.71 | 87.70 | 87.71 | 87.70 | 87.71 | 87.70 | 87.71 | 87.70 | 87.71 | 87.70 |
| WeCFL | 88.35 | **79.86** | **88.53** | 79.42 | 88.31 | 78.75 | 88.02 | 78.43 | 88.34 | 78.43 |
| BCFL-G | 87.68 | 80.03 | 89.00 | 80.25 | 87.53 | **87.50** | 89.30 | 83.4 | 89.27 | 81.92 |
| BCFL-C-3 | 88.55 | 80.19 | **89.31** | 80.98 | 89.19 | 82.11 | 89.07 | **82.44** | 89.26 | 80.61 |
| BCFL-C-6 | 88.50 | 80.86 | 88.64 | 81.48 | 88.66 | 81.63 | 88.76 | **82.60** | **88.90** | 81.25 |
| BCFL-MH-3 | 88.66 | 81.47 | **89.56** | 81.38 | 89.04 | 82.68 | 89.08 | 82.96 | 89.24 | **83.25** |
| BCFL-MH-6 | 88.08 | 80.80 | **88.90** | 80.95 | 88.65 | **82.78** | 88.84 | 82.64 | 88.77 | 82.47 |

It can be seen that when the assumed $K$ is smaller than $4$, the results degrade to some degree for all methods, while not dramatically. On the other hand, when the cluster number is bigger than $4$, the results are usually very similar to the results under the correct $K$ specification. A large number of assumed clusters is therefore desirable from a performance perspective, although it comes at a higher computational cost. A small number of clusters usually means less computation cost, but may reduce the personalized features of BCFL. In the open literature, related works refer to this challenge. Some suggest choosing the number of clusters by running a small experiment with a few rounds (Long et al., 2022), while others present methods found in most clustering algorithms such as the information criterion approach (Goutte et al., 2001; Sugar & James, 2003), which are not seen in FL clustering. This may raise great interest in how to apply them, as well as how to learn $K$ also from the perspective of data association theory that drives our work.

Table 10: Table 1 with standard deviation

| Datasets | Digits-Five | | AmazonReview | | Fashion-MNIST | | CIFAR-10 | |
|---|---|---|---|---|---|---|---|---|
| non-IID setting | Feature $(5, 2)$ | | Feature $(4, 2)$ | | Label $(4, 10, 0.1)$ | | Label $(4, 10, 0.1)$ | |
| Methods | Acc | F1 | Acc | F1 | Acc | F1 | Acc | F1 |
| FedAvg | $93.18 \pm 0.02$ | $92.95 \pm 0.02$ | $87.71 \pm 0.18$ | $87.70 \pm 0.18$ | $85.00 \pm 0.24$ | $53.14 \pm 0.32$ | $39.89 \pm 0.18$ | $21.19 \pm 0.18$ |
| WeCFL | $93.81 \pm 0.20$ | $93.58 \pm 0.20$ | $85.57 \pm 0.33$ | $85.51 \pm 0.34$ | $96.74 \pm 0.10$ | $92.0 \pm 0.50$ | $72.91 \pm 0.25$ | $51.69 \pm 0.41$ |
| BCFL-G | $94.35 \pm 0.00$ | $94.16 \pm 0.00$ | $87.53 \pm 0.00$ | $87.50 \pm 0.00$ | $96.81 \pm 0.06$ | $93.51 \pm 0.92$ | $64.73 \pm 7.74$ | $44.22 \pm 7.00$ |
| BCFL-G-W | $94.48 \pm 0.00$ | $94.29 \pm 0.00$ | $87.58 \pm 0.00$ | $87.57 \pm 0.00$ | $96.69 \pm 0.05$ | $92.21 \pm 0.15$ | $72.95 \pm 0.29$ | $52.01 \pm 0.64$ |
| BCFL-C-3 | $94.04 \pm 0.11$ | $93.84 \pm 0.10$ | $87.95 \pm 0.18$ | $87.93 \pm 0.18$ | $96.84 \pm 0.02$ | $90.16 \pm 1.30$ | $72.12 \pm 0.21$ | $50.97 \pm 0.51$ |
| BCFL-C-3-W | $94.35 \pm 0.05$ | $94.17 \pm 0.04$ | $88.22 \pm 0.26$ | $88.20 \pm 0.26$ | $96.88 \pm 0.07$ | $90.06 \pm 1.12$ | $73.18 \pm 0.29$ | $52.61 \pm 0.41$ |
| BCFL-C-6 | $94.14 \pm 0.11$ | $93.93 \pm 0.12$ | $88.11 \pm 0.42$ | $88.09 \pm 0.43$ | $96.58 \pm 1.60$ | $86.55 \pm 1.24$ | $68.73 \pm 1.11$ | $47.40 \pm 1.29$ |
| BCFL-C-6-W | $94.42 \pm 0.02$ | $94.24 \pm 0.02$ | $87.96 \pm 0.42$ | $87.95 \pm 0.43$ | $96.71 \pm 0.04$ | $87.31 \pm 2.45$ | $72.22 \pm 0.14$ | $50.02 \pm 0.47$ |
| BCFL-MH-3 | $95.39 \pm 0.32$ | $95.22 \pm 0.35$ | **88.74** $\pm 0.19$ | **88.74** $\pm 0.19$ | $97.13 \pm 0.33$ | **93.70** $\pm 3.77$ | $74.35 \pm 0.98$ | $56.24 \pm 2.69$ |
| BCFL-MH-3-W | $95.83 \pm 0.08$ | $95.70 \pm 0.09$ | $88.38 \pm 0.26$ | $88.38 \pm 0.26$ | **97.40** $\pm 0.14$ | $93.42 \pm 0.83$ | $76.19 \pm 0.33$ | $58.42 \pm 0.40$ |
| BCFL-MH-6 | $96.02 \pm 0.00$ | $95.88 \pm 0.00$ | $88.16 \pm 0.23$ | $88.15 \pm 0.23$ | $97.27 \pm 0.00$ | $92.56 \pm 0.00$ | $75.26 \pm 1.75$ | $53.45 \pm 5.23$ |
| BCFL-MH-6-W | **96.22** $\pm 0.00$ | **96.08** $\pm 0.00$ | $88.43 \pm 0.18$ | $88.43 \pm 0.18$ | $97.33 \pm 0.07$ | $90.62 \pm 1.01$ | **77.56** $\pm 0.22$ | **59.18** $\pm 0.49$ |

In addition to Table 7, we provide additional results in this appendix. Notice that the analysis of all the results across the four experiments yield similar conclusions as discussed in Section 5.1 in terms of the superiority of BCFL, the ranking of the approximate solutions, as well as the ability to handle clustering hypotheses and associated uncertainty. This is a brief summary of those results:

Figure 5: Digits-Five (a) Accuracy (b) F-1 score (c) Accuracy warm-up comparison (d) F-1 score warm-up comparison.

- Figure 5 shows the Digits-Five accuracy and F1-score results. Comparing both with/without warm-up initializations, we can see that the effect of inclNding it is not dramatic.

- Figure 6 shows the Digits-Five clustering results. We can observe very consistent clustering of clients observing similar data, as well as cross-pollination from other clients in BCFL schemes which helps improve the model training.

- Figure 7 shows the AmazonReview accuracy and F1-score results.

- Figure 8 shows the AmazonReview clustering results. AmazonReview data is text data containing user's reviews from a variety of products. Since the data distribution across clients is not extremely different, we can observe that even WeCFL connects more clients and therefore shares more information in the training phase. In this case, the method cannot easily cluster the data into four groups. In the case of BCFL, although it becomes more challenging, we can observe certain patterns and stronger associations among clients that should be clustered together.

- Figure 9 shows the Fashion-MNIST accuracy and F1-score results.

- Figure 10 shows the clustering results for Fashion-MNIST. Most of the time, the clients are correctly grouped into 4 clusters, but they can also be grouped differently, especially with the proposed BCFL methods. This flexibility allows them to share more information with each other. Warm-up models are less shareable compared to models without warm-up, as a good initialization can lead to faster local convergence.

- Figure 11 shows the CIFAR-10 accuracy and F1-score results.

- Figure 12 shows the CIFAR-10 clustering results, yielding similar conclusions as with the Fashion-MNIST dataset.

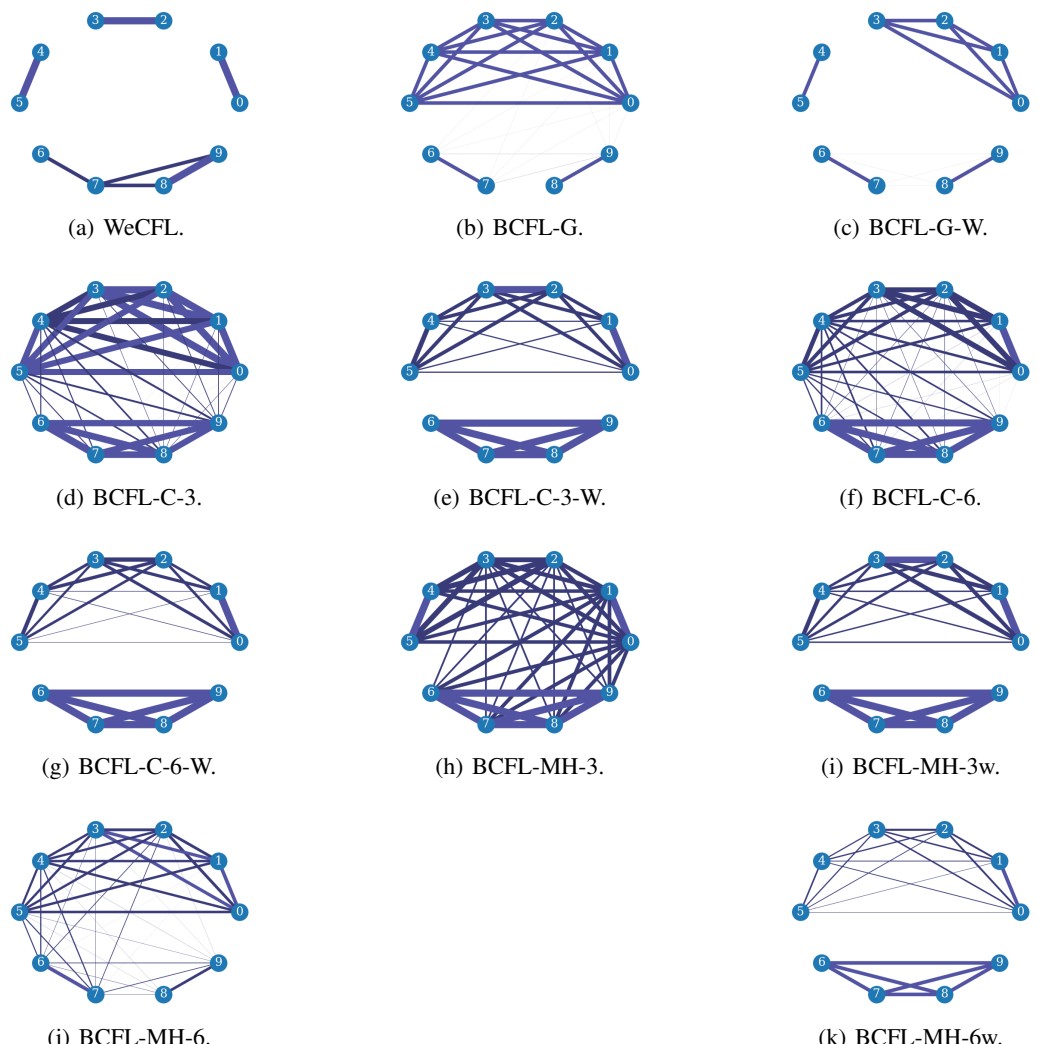

Figure 6: Client clustering during training for Digits-Five dataset. The experiment is such that of $K = 5$ data groups with $C/K = 2$ clients per group. It can be observed that the pairs of clients are grouped together, sometimes associated with other pairs as well.

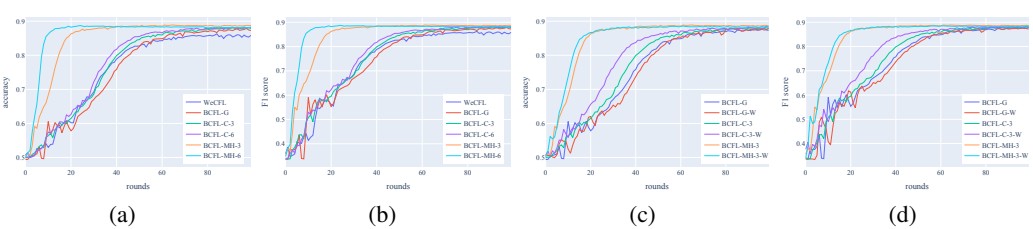

Figure 7: AmazonReview (a) Accuracy (b) F-1 score (c) Accuracy warm-up comparison (d) F-1 score warm-up comparison

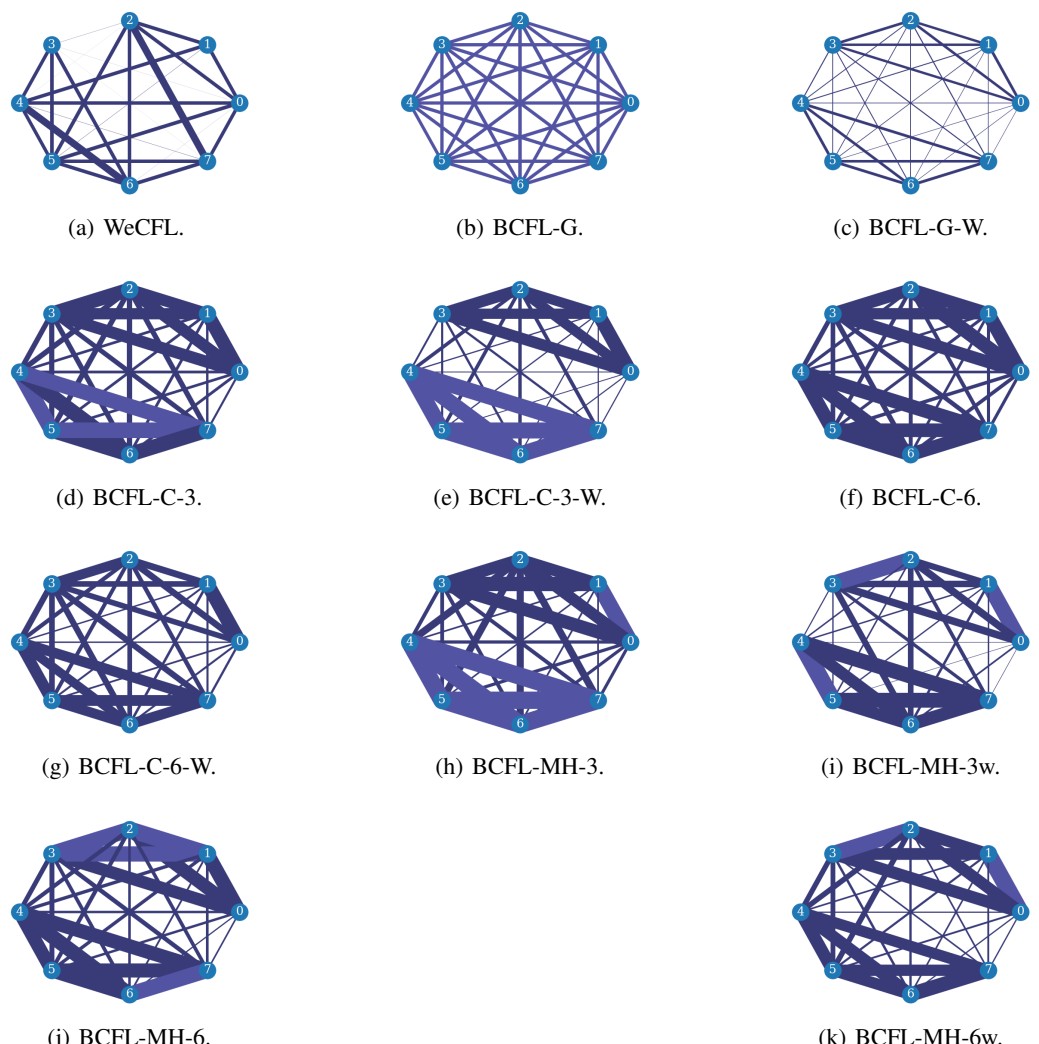

Figure 8: Client clustering during training for AmazonReview dataset. The experiment is such that of $K = 4$ data groups with $C/K = 2$ clients per group. It can be observed that the pairs of clients are grouped together, sometimes associated with other pairs as well.

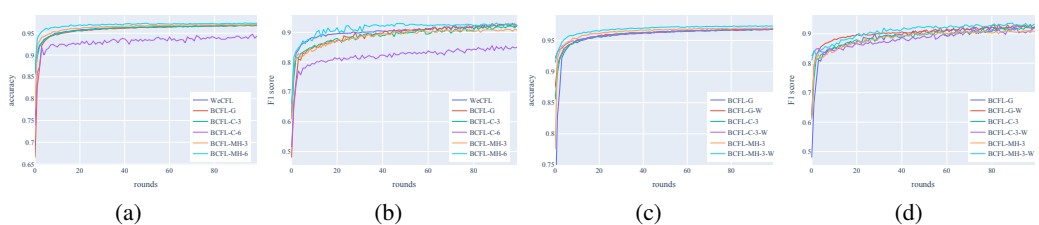

Figure 9: Fashion-MNIST (a) Accuracy (b) F-1 score (c) Accuracy warm-up comparison (d) F-1 score warm-up comparison

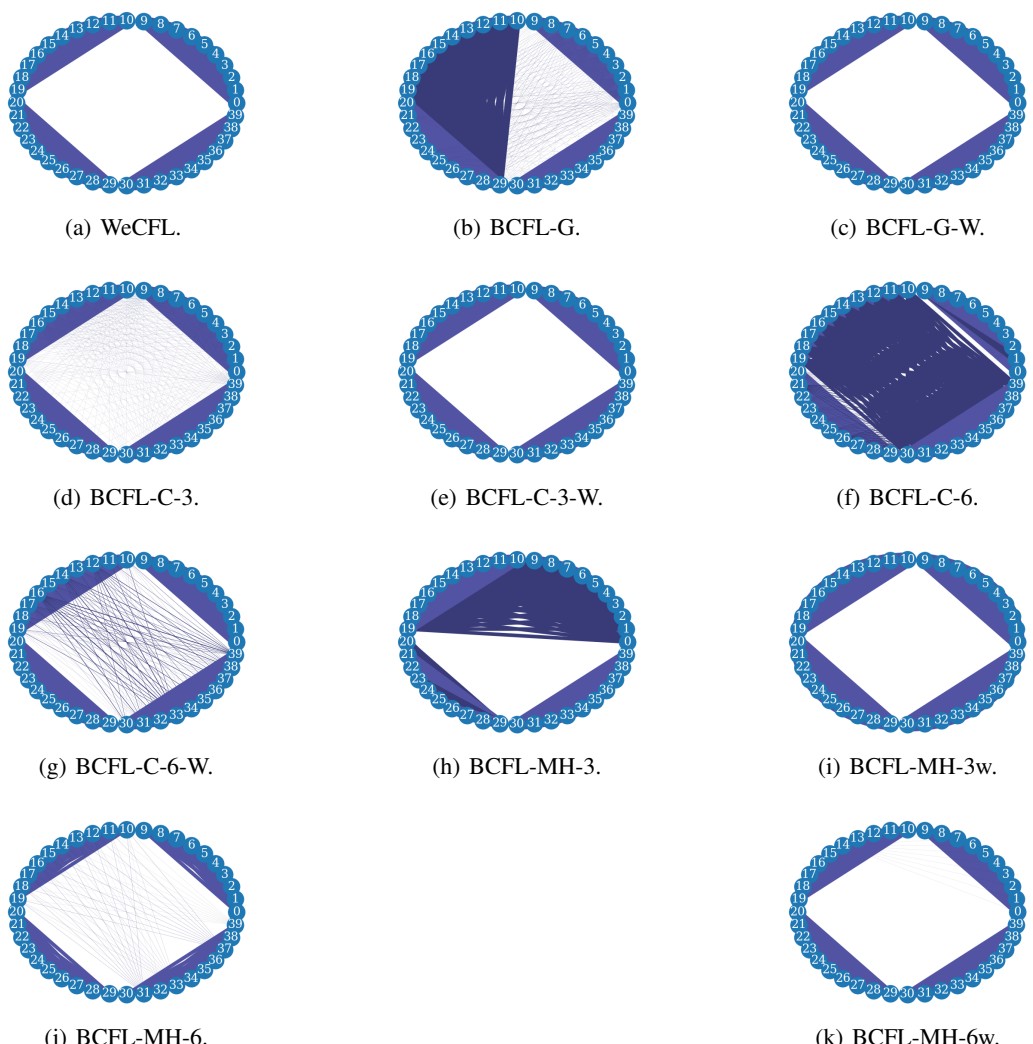

Figure 10: Client clustering during training for Fashion-MNIST dataset. Most of the time, clients are clustered in 4 groups, which is consistent with the experiment setup of $K = 4$ data groups with $C/K = 10$ clients per group.

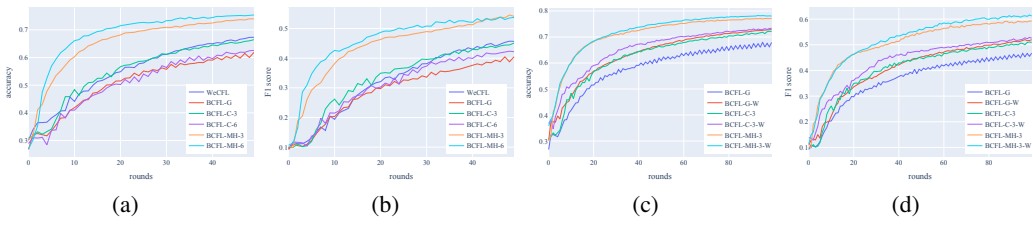

Figure 11: Fashion-MNIST (a) Accuracy (b) F-1 score (c) Accuracy warm-up comparison (d) F-1 score warm-up comparison

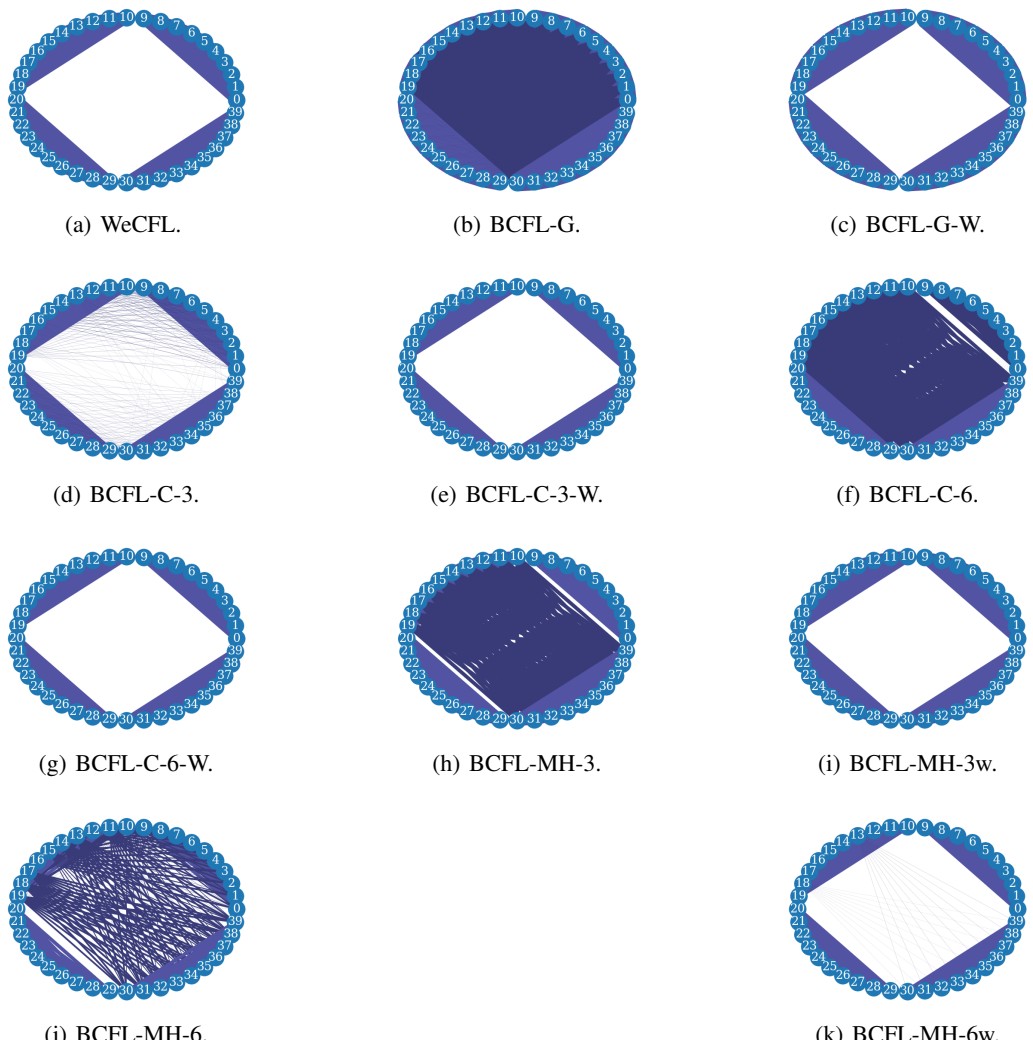

Figure 12: Client clustering during training for CIFAR-10 dataset. Most of the time, clients are clustered in $4$ groups, which is consistent with the experiment setup of $K = 4$ data groups with $C/K = 10$ clients per group.

