# OpenReview forum: "A Bayesian Framework for Clustered Federated Learning"
_ICLR.cc/2024/Conference — ICLR 2024 Conference Withdrawn Submission_

### Official Review · Reviewer_n2mn · 2023-11-01

**Soundness:** 2 fair
**Presentation:** 1 poor
**Contribution:** 2 fair
**Rating:** 5
**Confidence:** 2

**Summary:**

This paper considers the problem of clustered federated learning. Inspired by a Bayesian modeling of clustered FL, the provides three clustering heuristics. The paper evaluates the proposed heuristics and shows that in practice they outperform  WeCFL, which is the state-of-the-art FL clustering method.

**Strengths:**

- The proposed clustering heuristics exhibit superior performance compared to the current state-of-the-art WeCFL in federated learning.
- The experimental results section is well-crafted, providing a thorough evaluation of the proposed methods.
- BCFL distinguishes itself from other clustering federated learning methods by dynamically adjusting clusters, offering a unique approach that departs from hierarchical clustering strategies.

**Weaknesses:**

- The paper's overall clarity is compromised due to unnecessarily complex and at times undefined notation, making it challenging to follow.
- The update mechanism for the parameters of the Bayesian model lacks explicit explanation, leaving a crucial aspect of the methodology unclear.
- The comparison and discussion of related work are superficial, lacking a fundamental exploration of how the proposed method differs from other clustering approaches.
- The claim that the paper provides a "unified framework with a rigorous formulation" and offers "new theoretical insights and algorithms" is disputed, as the paper fails to present any novel theoretical contributions.
- Section 3 contains derivations that are deemed obvious, contributing minimally to the understanding of the proposed approach.
- The content in Sections 3 and 4.1 communicates a message that, in my interpretation, appears rather straightforward. The outlined two-step iteration involves computing the "optimal" association $\theta^{*}_{t}$ given cluster weights $\Omega$ and updating cluster weights based on the current association rule. Furthermore, the solution presented in equation (16) suggests associating a client $j$ with a cluster $i$ that maximizes $\log(D^j|w^i)$. While these concepts are essential, the simplicity of the presented message does not seem commensurate with the extensive coverage given to these sections (four pages). The detailed explanation does not justify the space allocated, and as such, Sections 3 and 4.1 may be perceived as overextended for the relatively straightforward content they provide.
- In light of the preceding remark, it might be beneficial for the paper to discuss soft clustering approaches based on the EM algorithm.
- The proposed methods lack theoretical guarantees, relying solely on heuristics without a solid theoretical foundation.
- The success of the proposed method heavily depends on the higher computational cost of BCFL-MH, raising concerns about the practicality and efficiency of the cheaper variants (BCFL-G and BCFL-C). In fact,  BCFL-G and BCFL-C do not show a significant improvement over FedAvg and WeCFL; the improvement never exceeds $1$ p.p., and is often lower then $0.1$ p.p.
- Several minor issues, including inconsistent citation style, grammatical errors, and unclear notation choices, need attention for a more polished presentation:
     - The paper seems to use the wrong citation style. It refers to the authors when it should refer to the paper. For example, the sentence "FL allows collaborative model training without data sharing across clients, thus preserving their privacy McMahan et al. (2017)" should be "FL allows collaborative model training without data sharing across clients, thus preserving their privacy (McMahan et al., 2017). "
    - In Section 1, "the is a lack" -> "there is a lack".
    - What is the reason behind using $\mathbb{P}$ instead of $P$ in page 3?
    -In Page 4, "to denote the a particular" -> "to denote a particular".
     - In Page 7, "both feature- and label- skew" -> "both feature---and label---skew"

**Questions:**

- My understanding of the paper is that each iteration is split into two steps: compute the "optimal" association $\theta^{*}_{t}$ given the weights of the clusters $\Omega$, then, update the cluster weights given the current association rule. Could you please confirm or deny my interpretation?
- Regarding (13), it seems for me that the optimal solution would pick $A^{i j} = 1$, for $i \in \text{arg}\min_{i} L^{i, j}$. It translates in (16), to  associating the client $j$ to the cluster $i$ that justifies the best its data, i.e. the cluster $i$ such that $log(D^j|w^i)$ is maximal. Could you please confirm or deny my claim?

---

> ### Author Response · Authors · 2023-11-16
> **Response to weak points 1~7**
>
> We thank the reviewer and we reply to the points raised by the reviewer.
> - The notation complexity is due to the indexing of the multiple association hypotheses, being random finite sets themselves, which is at the core of the conceptual solution of the association problem brought by clustering. It would be helpful if the reviewer could point out notation issues so we could address them. Notice that there is a summary table with notation in the appendix and that we are revising it to make sure relevant variables are defined.
> - At every communication round we use the Laplace approximation approach as discussed in "Baseline models and system settings" in Section 5.
> - We agree with the reviewer that the discussion presented in the related work section could be improved. We are working on improving the discussion for the revised manuscript.
> - We strongly disagree with the reviewer on this point. We formulated the problem of clustered federated learning as a Bayesian data association problem between clients and model parameter distribution. The theoretical approach presented in Section 3 is completely novel in the FL literature: section 3.1 presents the conceptual formulation and solution to compute the joint posterior of model parameters and data association; section 3.2 extends 3.1 to recursive Bayesian updates and associations, which are implemented as FL iterations and yield to an optimal yet intractable solution as pointed out in section 3.3. Furthermore, the derivations are far from trivial, they are rigorous in the sense that they rely on consolidated Random Finite Set and Bayesian theories with few assumptions, no approximations or heuristics. The proposed approach is a unified framework since the theory proposed in Section 3 is general and can lead to multiple solutions as those in Section 4. The proposed conceptual solution, however, leads to a high number of data associations and, consequently, to high computational complexity if applied naively. Nevertheless, the insights provided by the theory laid out in Section 3 were used to propose the approximations discussed in Section 4 which successfully aim at reducing the complexity through novel algorithmic solutions. We modified the title of Section 3 as "Optimal Bayesian solution for clustered federated learning" to make it clear that the material there is already part of the novel contribution of this paper, which we agree was not clear earlier when titled "Problem formulation". Overall, we believe that the proposed approach touches all the points we claimed. We hope that the reviewer can reconsider such a strong claim that the paper fails to present any novel theoretical contribution.
> - We disagree, see our previous answer.
> - The reviewer is mistaken. Section 3 and 4.1 present different pieces of the proposed BCFL: while section 3 presents the conceptual solution where all association hypotheses are kept, section 4.1 presents the basics to weight different hypotheses and design approximate solutions to the computational challenge. The solution of equation (16) leads to the set of data associations that maximizes the sum, over all $K$ clusters and all clients in each cluster, of the $\log$ unnormalized marginal posterior of data association weights $\tilde{\pi}^{\theta^{ij}_t|h}$, where $h\triangleq \theta\_{1:t-1}$ indicates the history of past associations as discussed below equation (11).
> In the case where Equation (9) is approximated using the expected value of the model parameters then $\tilde{\pi}^{\theta^{ij}\_t|h} = p(\mathcal{D}^j|\omega^i\_{t-1})$. Nevertheless, only in the greedy approach discussed in Section 4.2 the selection is made solely on the $\log p(D^j|\omega^i\_{t-1})$.
> We hope that the reviewer can revisit this point and change the overall evaluation of the manuscript.
> - We will add these references to the related work section discussing it connects to BCFL. Note that a FedSoft is already cited in our work.  In our paper, we focused on the clustered FL problem as stated in (Ghosh et al., 2020; Mansour et al., 2020), where the objective is to learn multiple models under the assumption that data from each client comes from a specific distribution.
> Our approach focuses in clustering the different clients using a data association Bayesian framework that provides a joint posterior distribution of client associations and model parameters. Opposed to existing methods, we consider multiple data association hypotheses that enable the uncertainty quantification of client-to-cluster associations, leading to noticeable performance improvement.
> In EM-based FL approaches the assumption is that the data at each client belongs to a mixture of distributions. Then, FL approaches are used to independently train multiple models under this assumption, which does not explore different association hypotheses. Furthermore, FedEM still relies on frequentist estimation strategies which contrasts with our Bayesian framework.

---

> > ### Author Response · Authors · 2023-11-16
> > **Response to weak points 8~10 and Questions**
> >
> > - We disagree with this comment. Our proposed framework is optimal in the Bayesian sense, where the objective is the computation of the joint posterior distribution of the model parameters and the clustering hypotheses. The solutions in Section 4 are suboptimal, although they ultimately aim at approximating the joint posterior while maintaining tractability. The theoretical guarantees come from our choice of the Bayesian paradigm, which states that all information is contained in the full posterior distribution that we are estimating. Consequently, the claim that our results rely solely on heuristics without a solid foundation is false.
> > - The performance can vary depending on the dataset and algorithms used, but according to the results in the main body and those in the appendix all proposed methods show improvement.
> > The cheaper variants pointed out are comparable to WeCFL, which was expected due to the assumptions made by the methods and the way the clients were clustered in the experiment. However, they outperform WeCFL in more challenging clustered scenarios, such as that presented in Table 7 in the appendix for Fashion-MNIST and Amazon-Review datasets. Noticeably, all variants of the method outperform FedAvg results in general, sometimes with substantial improvements (Fashion-MNIST and Cifar10) and sometimes with marginal improvements (Digit-5 and Amazon-Review).
> > - We thank the reviewer for pointing those typos out. We fixed them in the revised manuscript.
> >
> > Question 1:
> > - Not an accurate claim in general since this only explains BCFL-G in Section 4.2. In the conceptual solution of Section 3 the model is updated under all association hypotheses, while in the practical algorithms in Section 4 only a subset of hypotheses are considered: BCFL-G updates the "best" hypothesis, as the reviewer points out;  BCFL-C updates a set of $M$ "best" hypotheses then fuses them to upload a single model; and BCFL-MH updates multiple hypotheses and uploads them to the server, where merging/pruning happens to approximate the posterior while limiting the number of hypotheses at every learning round.
> >
> > Question 2:
> > - This is not an accurate claim in general, because it again explains the BCFL-G variant. Notice that the optimization in (13) requires the $\tilde{\pi}^{\theta^{ij}_t|h}$ from all clients, which are then jointly used to rank the best hypotheses in BCFL-C and BCFL-MH variants.

---

> > > ### Comment · Reviewer_n2mn · 2023-11-27
> > >
> > > I want to thank the authors for their effort to address my questions and concerns, and I apologize for the delay in providing this feedback. I have carefully re-read the paper and the authors' rebuttal, and my short feedback is: that the paper provides nice contributions, but I am not 100% sure if the paper is above the acceptance threshold.
> > >
> > > I remind you that I appreciate the contributions of the paper, notably the experimental results and the ability of BCFL to dynamically adjust clusters. My main concerns are: 1) the mathematical derivations are mostly straight-forward and often are immediate applications of the Bayes rule, and 2) the proposed algorithms (BCFL-G, BCFL-C, and BCFL-MH) are heuristics, in the sense that they do not have formal guarantees.
> > >
> > > Please find below-detailed feedback on each of the points addressed by the rebuttal:
> > >
> > > * Regarding the question regarding notation, I do not see why it is necessary to use random sets to model the problem. A simple alternative is to include a random variable that represents if a client is assigned to a client. Additionally, I do not see why we need to define $\pi^{\theta}, p^{\theta},\dots$. These quantities hide the dependencies on the parameters of the graphical model.
> > >
> > > * Can you, please, point out exactly what are the "novel theoretical contributions"? From what I can see, the paper models the problem of clustered federated learning using random sets and obtains an expression of the posterior distribution using the Bayes rule.
> > >
> > > * I agree with the authors that the derivations are rigorous; I did not claim the converse in my review. My claim is that the derivations are trivial; in fact, almost all derivations are immediate applications of the Bayes rule. Can you, please, point out what step of the derivations is challenging to obtain?
> > >
> > > * Regarding the use of the expression "unified framework" is that it can capture other\most clustered federated learning. methods. The rebuttal suggests that it is in the sense that it can lead to multiple heuristics, i.e., the heuristic in Section 4. I suggest the authors clarify the meaning of "unified" to eliminate any ambiguity.
> > >
> > > * I agree with the authors' rebuttal regarding the difference between Section 4.1 and Section 3. However, this point is not critical in the overall evaluation of the paper.
> > >
> > > * The paper derives the posterior update rule. However, "the proposed conceptual solution,.., leads to a high number of data associations and, consequently, to high computational complexity. "The paper "design[s] approximate solutions to the computational challenge." The paper does not provide theoretical guarantees for the approximate solutions; hence my claim that the paper relies on heuristics.

---

> > > > ### Author Response · Authors · 2023-12-02
> > > > **General reply and point 1~2**
> > > >
> > > > We thank the reviewer for the time and effort spent into this revision process, as well as the appreciation of our work. We hope the clarity has improved and that the relevance of its contributions is better positioned. In our perspective, the paper provides a new principled methodology that can lead to many different approximate solutions that can go far beyond what we achieved in Section 4 of the manuscript. It provides top-notch results (better than the state-of-the-art) and we are worked on clarifying the algorithm pipeline (adding pseudo-code descriptions in the Appendix). We believe the contribution is substantial and that this work is a great fit for ICLR audience.
> > > >
> > > > 1) We call the reviewer's attention to the fact that the conceptual ideas motivating those derivations are not trivial, and different aspects of the formulation were performed to either, result in a more tractable problem or to connect with the distributed processing paradigm employed in FL.
> > > >  For instance, the fact that we are not computing the posterior as in Equation (1) but as in Equation (2), allows us to: $(i)$ develop the distributed Bayesian framework which is discussed in Equations (5)-(9); and $(ii)$ allows for a simpler recursive strategy that is presented in Section 3.2.
> > > >  We highlight that all those steps were carefully thought through and conceptualized to cope with the Bayesian data association and the CFL paradigms, bringing challenges regarding its  motivation, design, and conceptualization.
> > > >  For instance, in Equations (2)-(9) intricate application of Bayes Theorem, marginalization, product rules, and integral approximation were leveraged leading to a posterior formulation that connects with the FL paradigm.
> > > >  For these reasons, we believe that the derivations presented in Section 3.1, despite their algebraic simplicity, are not trivial and show the chain of thought applied to the problem.
> > > > Of course, a Bayesian framework will rely on the usage of the Bayesian Theorem in its derivations.
> > > >
> > > > 2) Note that the conceptual solution has the optimality guarantees of the Bayesian methodology providing the full posterior characterization of associations and parameters. We understand the reviewer's point regarding the approximations performed in the practical implementations. These are indeed suboptimal since not all hypotheses are explored. We acknowledge the reviewer's concern regarding the absence of a formal guarantee in this part, or at least guarantees in the sense of those provided in frequentist-based methods. As a matter of fact, our proposed approximation BCFL-G has strong connections to the IFCA [ghosh2020efficient], when formulated in a Bayesian context, which provides those theoretical guarantees. Our paper concentrates on the Bayesian framework rather than on particular deterministic algorithms, which is why we did not initially include a convergence proof. However, we recognize the importance of such an analysis, and we are certainly open to investigating convergence guarantees in future research.
> > > >
> > > > Please find below-detailed feedback on each of the points addressed by the rebuttal:
> > > >
> > > > 1. The reviewer is correct in that other modeling approaches could be taken. The approach proposed by the reviewer could be valid when the number of clusters are known and assumptions limiting the association between clients and clusters are performed. However, we wanted to introduce a general framework in Section 3.1 that could allow for arbitrary associations between clients and clusters.
> > > > In the practical implementations discussed in Section 4 we assumed prior information about these quantities, and that a client can only be associated with a single cluster. However, the opportunity for novel contributions is presented, which we will explore in upcoming work.
> > > >
> > > > 2. This is not a mere application of the Bayes rule, as the reviewer seems to suggest. As detailed above on our first response, there are many steps that are aimed at obtaining 1) a recursive solution to update the join posterior of parameters and associations, and 2) a distributed version where relevant quantities (e.g. association weights) can be computed locally thus enabling a FL scheme. By approaching the paper from a data association perspective we make this possible. Of course, a Bayesian framework will rely on the usage of the Bayesian Theorem in its derivations, so the reviewer's comment is disconcerting. The novel theoretical contribution in this work is the connection of data association theory to CFL problem. We improved the related work section to better position our work, for which we hope that this connection is made more clear to the reviewer.

---

> > > > > ### Author Response · Authors · 2023-12-02
> > > > > **Point 3 ~ 6**
> > > > >
> > > > > 3. We call the reviewer's attention to the fact that the conceptual ideas motivating those derivations are not trivial, and different aspects of the formulation were performed to either, result in a more tractable problem or to connect with the distributed processing paradigm employed in FL.
> > > > >  For instance, the fact that we are not computing the posterior as in Equation (1) but as in Equation (2), allows us to: $(i)$ develop the distributed Bayesian framework which is discussed in Equations (5)-(9); and $(ii)$ allows for a simpler recursive strategy that is presented in Section 3.2.
> > > > >  We highlight that all those steps were carefully thought through and conceptualized to cope with the Bayesian data association and the CFL paradigms, bringing challenges regarding its  motivation, design, and conceptualization.
> > > > >  For instance, in Equations (2)-(9) intricate application of Bayes Theorem, marginalization, product rules, and integral approximation were leveraged leading to a posterior formulation that connects with the FL paradigm.
> > > > >  For these reasons, we believe that the derivations presented in Section 3.1, despite their algebraic simplicity, are not trivial and show the chain of thought applied to the problem. Of course, a Bayesian framework will rely on the usage of the Bayesian Theorem in its derivations.
> > > > > As mentioned previously, we first aim at the reader understanding how data association theory relates to CFL problems, despite its basis in Bayes theory. While Bayes theory may be fundamental and straightforward, its significance lies in its application. In our work, we introduce hypotheses or associations when derived from the Bayesian rule specifically for the CFL problem. This approach provides a solid derivation as the reviewer also said it's from the Bayesian rule.
> > > > >
> > > > > 4. Good point. We updated the related work section to discuss data association related works, for which there is extensive research in general non-FL problems. We also updated the paper to clarify the `unified framework' terminology, through which we mean that some existing works in CFL can be interpreted as variants of the presented BCFL. That is, particular practical schemes as those presented in Section 4. For instance, the popular IFCA ghosh2020efficient] has strong connections to BCFL-G when taking a purely frequentist (e.g. non-Bayesian) approach.
> > > > > This reinterpretation of CFL using data association and Bayesian modeling creates opportunities to apply existing methods from those fields to the CFL problem and even to develop new methods suited for this area.
> > > > >
> > > > > 5. We thank the reviewer for the agreement.
> > > > >
> > > > > 6. We disagree with the reviewer when they qualify the approximations provided in Section 4.1 as heuristics, although it might be a matter of semantics. In our understanding, the conceptual BCFL solution in Section 3 is optimal (in the Bayesian sense, where all necessary information is gathered in the posterior distribution, which is evaluated without approximations). Then, due to the unfeasibility of such solution, Section 4 proposes three different practical algorithms. These are suboptimal (or heuristics, in the reviewer's terms) as is agreed and expressed throughout the paper. That being said, these approximations are not arbitrary (maybe this is where our semantic misunderstanding is) but they approximate the optimal solution under certain computational constraints. The better those approximations are, the closest to optimal they are. However, we acknowledge the reviewer's concern regarding the absence of a formal guarantee in this part, or at least guarantees in the sense of those provided in frequentist-based methods. As a matter of fact, our proposed approximation BCFL-G has strong connections to the IFCA [ghosh2020efficient], when formulated in a Bayesian context, which provides those theoretical guarantees. Our paper concentrates on the Bayesian framework rather than on particular deterministic algorithms, which is why we did not initially include a convergence proof. However, we recognize the importance of such an analysis, and we are certainly open to investigating convergence guarantees in future research.

---

### Official Review · Reviewer_Yz5m · 2023-11-05

**Soundness:** 2 fair
**Presentation:** 2 fair
**Contribution:** 2 fair
**Rating:** 5
**Confidence:** 3

**Summary:**

This paper proposes a clustered FL method based on a bayesian framework which assigns clients to clusters, and provides three different variations of the proposed bayesian clustered FL framework to address practical considerations, namely approximate BCFL, greedy BCFL, and consensus BCFL. Specifically, for $K$ clusters and $C$ clients, BCFL assigns each client to its optimal cluster based on a target posterior characterization. The work performs preliminary experiments to validate the BCFL's performance on CIFAR10 and FMNIST.

**Strengths:**

- The work proposes many different variants of BCFL that considers practicality.
- The work investigates clustering in FL dependent on the different data distributions of the clients which is a relevant topic in FL where data heterogeneity can be particularly severe.
- The work includes details of their experimental results including graphs that show the relationship across clients and their relatedness with the results.

**Weaknesses:**

- A major concern I have is regarding the efficacy and practicality of the proposed framework. Although the authors have proposed the three variants of the proposed BCFL framework, they still require the downloading of the models and weights under past associations, and then again uploading the associations of the weights to get the association decisions again from the server. Then finally the clients upload the local models based on the conditional associations. This requires at least 2 times the communication rounds compared to the conventional FL framework as well as more computation imposed to the clients. I became more skeptical after looking at the experimental results which only include 10 clients in total or 8 clients in total for more cross-device like datasets such as digits-dive or amazon review. Another concern regarding this approach of BCFL is the sensitivity of the number of clusters $K$ to the performance. It will be difficult to know this value in advance in practice. How does the authors address this problem as well? Overall due to these issues I am skeptical of the efficacy and practicality of the proposed framework for realistic FL settings.

- Another concern I had regarding the practical variant approximate BCFL, the authors assume that there is no overlap in the distribution across different client partitions. This is quite a strong assumption which does not hold in most of the cases in FL. Can the authors comment on this assumption and how realistic it is?

- The writing of the paper can be improved. for instance in pg1 when the authors address problem 1 and problem 2 in bold, there seems to be a typo/error. Ex: Problems2: The is a lack of a united theory. Moreover, optimality is used throughout the paper from the beginning without a proper explanation on what the authors exactly mean by this. It can mean differently for different readers. In addition, the presentation of Figure 3 can be improved, It is quite hard to see the differences across the curves.

Due to these concerns, I am leaning towards rejection for the work.

**Questions:**

See weaknesses above.

---

> ### Author Response · Authors · 2023-11-16
>
> We thank the reviewer for the appreciation of the strengths of our work and the feedback provided. Below we reply point by point to the weaknesses perceived by the reviewer.
> -  There are three comments within the first item. Namely:
> 1) It is worth clarifying that downloading the model for all past hypotheses is only the case for the conceptual solution discussed in Section 3 of the paper. However, the approximate solutions proposed in Section 4 are aimed to reduce those communication needs and thus to improve the practicality of BCFL. Other things to consider are that i) the weight associations variable sent along the model weights are just a few scalar values, thus requiring minimal extra communication cost; and ii) this paper tackles the additional communication cost by way of the proposed approaches that reduce the need to send an otherwise growing number of hypotheses.
> 2) We also conducted tests involving a larger number of clients and have presented additional examples with 40 clients in the appendix, we will emphasize this in the updated paper. In the main body, we included only a limited number of clients for the Digit5 and Amazon Review datasets to ensure the clarity of our cluster analysis results, as done in other related works. As depicted in Figure 4, the behavior of the clusters during training is distinctly observable.
> 3) Regarding the knowledge of the number of clusters $K$, we preliminary followed common practice in CFL works where this is often predetermined. We agree that estimating $K$ is of interest and it is actually on our future research plan, which our Bayesian framework can account for although not straightforwardly enough to be included in this paper so we instead decided to keep it fixed (although not necessarily correctly specified). To address this comment we updated the appendix with an experiment where the sensitivity to under- and over-estimating $K$ is discussed.
> - We would like to clarify that the main assumptions in the paper are that given an hypothesis the model parameters per cluster are independent and that the local datasets are conditionally independent. The intuition behind the validity of those assumptions comes from the conditioning on the clustering hypothesis, whereby if we were to explore all of the hypotheses we would find that different clusters obey different models. We agree that this might not be always true, that is the reason behind explicitly stating it as an assumption. Doing so, we enable tractability of the problem, which otherwise would be extremely challenging. Finally, it is worth noting that other non-Bayesian CFL works implicitly make this assumption, although since those works are often not probabilistic this might not be as apparent as in our framework.
> - Thanks for pointing out the typos and figure suggestions, they are fixed. As for the use of `optimality', we refer to the fact that the conceptual solution in Section 3 provides the full posterior distribution of the model, where all association hypotheses are explored. That is, BCFL is optimal in the Bayesian sense, where the objective is the computation of the joint posterior distribution of the model parameters and the clustering hypotheses. Optimality of the algorithms is thus referring to a Bayesian inference notion of optimality, which is now mentioned in the first paragraph of Section 3. The approximate solutions developed in Section 4 are therefore suboptimal, although they ultimately aim at approximating that same joint posterior while maintaining tractability and practicality of BCFL.

---

### Official Review · Reviewer_UPyv · 2023-11-27

**Soundness:** 3 good
**Presentation:** 1 poor
**Contribution:** 3 good
**Rating:** 5
**Confidence:** 3

**Summary:**

The paper essentially extends the idea of model-based clustering under the Bayesian framework into the clustered federated learning (CFL) problem, where not only the partition of clients but also the model update through federated learning is considered. The paper proposes to tackle the exploded data association incurred by the recursive updates with the $M$-best assignments problem solver.

**Strengths:**

1. The paper's theoretical contribution that systematically discusses CFL under the Bayesian framework could serve broader interests apart from the proposed algorithms.

2. The efficiency problem in the recursive update could be tackled with the M-best assignment solver when using the negative log weights as the cost function.

**Weaknesses:**

1. ***Presentation***

There is a clarity problem before section 4. Specifically, the full procedure of the learning is poorly illustrated. Though Figure 1 is supposed to illustrate the essential steps and communications, the confusing coloring and the lack of clarity on the calculation of key values pose unnecessary challenges for the audience. The pseudo-code for the general learning, clustering procedure, and each variant of association selection, along with a detailed congregate discussion of the communication, would greatly improve the clarity of the paper. Also, it might help to focus on the conceptual discussion in Figure 1 rather than placing notations, which are introduced later in sections 3 and 4.

2. ***Related work***

Section 2 mainly focuses on discussing federated learning works. Yet, the paper needs to pay attention to the existing product partition models and needs more discussion over the progression of existing Bayesian frameworks, which could better position the paper's theoretical contribution.


3. ***Method***

The paper proposes to leverage the cost function to transform the association selection into the M-best assignments problem. Yet, as admitted in the discussion of section 4, the greedy or top $M$ trajectory could be myopic. This problem seems intrinsic in the definition of $L$ and is reflected in the empirical results where the BCFL-MF-6 does not constantly improve upon BCFL-MF-3.

4. ***Experiment***

(1) The datasets tested in section 5 seem to be manually synthesized, raising questions about the motivation of CFL and undermining the significance of the experimental results.

(2) The results only show the downstream performance while lacking evidence about the efficiency, especially for different approximations and corresponding key hyperparameters, including the $M$ discussed in sections 4.3 and 4.4 and $K$ that could potentially impact the communication efficiency and learning convergency.

5. ***Minors***

(1) The paper claims the existing CFL methods do not effectively exploit similarities between different clusters, yet needs more detailed discussion over the statement.

(2) The adopted merge operator in section 4.3 needs to be clarified.

(3) The model warm-up relies on a different partitioning method based on the local model initialization. And it is unclear why it contributes to the performance.

**Questions:**

The paper has an intense layout but lacks details of the algorithm design, e.g., a concrete pipeline of the algorithm and congregated discussion of the details of the communication between clients and servers corresponding to the proposed three variants. Based on the discussion and the experimental results, it seems that BCFL-MH is the best option among all the three approximations. If so, why not propose BCFL-MH as the main algorithm, allowing a better illustration of the algorithm's key concepts?  Is there any additional trade-off on the efficiency and performance when choosing the approximation?

---

> ### Author Response · Authors · 2023-12-01
>
> We thank the Reviewer for the appreciation of the great summary and strengths of our work by pointing out the contribution related to data association and Federated Learning.
>
> 1. Presentation:
>
> We improved the notation readability for those sections, as well as added pseudo-code algorithmic description for both the conceptual solution (section 3) and the practical schemes (section 4). These are in Section B of the Appendix.
> We acknowledge the possibility of enhancing these sections for clarity. However, given the constraints of time, it is challenging to undertake substantial revisions in less than a day. Nevertheless, we are committed to addressing the reviewer's concerns to the best of our ability within the time available and in future iterations.
>
> 2. Related work:
>
> Thanks for the pointers regarding how to improve the related work section. We improved it by adding more references and better motivating the contribution of this work within those works. Particularly, we discussed Bayesian frameworks within FL, as the reviewer suggested, which has indeed improved the positioning of this paper.
>
> 3. Method:
>
> Thanks, as the reviewer points out, the 3 methods proposed in the work are approximated because we cannot account for all the associations (discussed in section 3.3). More precisely: BCFL-G only keeps the best association every time; BCFL-C chooses the $M$ best associations at a given time instant and merges them into a single posterior solution, thus only propagating one hypothesis to the next communication round; and BCFL-MH keeps the $M$ best associations onto the next recursively update. There might be other approaches to approximating the full posterior, which itself can lead to different BCFL algorithms. Indeed, the results for BCFL-MH-6 do not always improve those from BCFL-MH-3. There could be multiple reasons to that such as related to the dataset and its setting, the chosen associations, and the randomness of each trial. Ultimately, when taking a close look at the result, MH-3 or MH-6 are noticeably close to each other, which indicates that for those experiments going from $M=3$ to $M=6$ does not provide substantial performance improvement.
>
> 4. Experiment:
> - All these datasets are real datasets. The usage of those datasets in the context of non-IID challenges is according to similar CFL works, which we compare to. We do not understand which questions regarding the motivation of CFL the reviewer is referring to. More details about the datasets are provided in the Appendix.
> - Per the reviewer's suggestion, we included a discussion on the communication efficiency for each practical algorithm in Appendix B.
>
> 5. Minors:
>
> (1) We provide a bit more discussion of the claim in the `Problem 1' description in the introduction section. The statement regarding other CFL methods is that the knowledge of each participating client is exploited by only one cluster during each round, which potentially results in an inefficient utilization of the local information that could contribute to the training of multiple clusters instead;
>
> (2) We improved it and more details can be found in a newly added Appendix C, as well as the references therein;
>
> (3) We implemented the warm-up method to see if it could help achieve faster convergence. And indeed, it did help. With the same number of communication rounds, it usually achieves better results. The reason behind this is that we provide a good initial estimate at the beginning, as shown in Figure 3. Warm-up methods typically have good initial accuracy and are used in other works.
>
> Questions:
>
> We improved the notation readability for those sections, as well as added pseudo-code algorithmic description for both the conceptual solution (section 3) and the practical schemes (section 4).
> These are in Section B of the Appendix. These pipeline descriptions should help improve the presentation of the work, as well as the information flow in the different methods (including the conceptual solution.
>
> Regarding the option of simply presenting BCFL-MH, our main objective is to present the BCFL framework from which one can derive the three proposed solution and many others that can be explored in future research from us or other groups.
> The objective of this work is to present a new framework for the problem of clustered FL, rather than just presenting a detailed method. The three approaches are practical implementations, derived from the framework. There could be many other variations by pruning, merging, and combining them, which would expand the possibilities to make progress in the CFL field. The BCFL-MH approach yields better results because it considers more associations compared to the other approaches. However, it is not as efficient as others, as discussed in the communication section of the appendix. That is the reason why we presented the three alternatives, balancing communication needs and performance. Each application might have different requirements.

---

> > ### Comment · Reviewer_UPyv · 2023-12-02
> >
> > I appreciate the effort of the authors to address my concerns. My remaining concerns for this submission are the following.
> >
> > (1) As also pointed out by my fellow reviewer, the current presentation on the analysis and methods poses a severe challenge to the audience to parse and lacks clarity, which, in some cases, could be misleading. I appreciate the author's effort in adding the additional discussion and pseudo-code in the appendix, yet a substantial revision of the main paper to address the clarity issue might be necessary.
> >
> > (2) As acknowledged by the author and in additional discussion in the appendix, there are trade-offs in choosing the practical algorithms apart from the reported downstream performance in the main paper. The missing trade-offs are critical to the discussion about the three algorithms. Suppose the author insists on containing all three methods. In that case, a more thorough discussion of the trade-off, including an empirical comparison of the communication and computation cost among the proposed three methods and with SOTA baselines, might be desired in the main paper.
> >
> > The following are clarifications and additional comments.
> >
> > (1) Essentially, the BCFL-G, BCFL-C, and BCFL-MH could be regarded as the same algorithm relying on the M-best assignment subroutine as heuristic to tackle the intractability. It's mostly about choosing different values for $M$ and whether to keep the multiple hypotheses or merge by introducing additional heuristics. I don't see the necessity to present the algorithms separately, which takes up much space and undermines conciseness.
> >
> > (2) My previous concern over the experiment is that the non-IID scenarios are not intrinsic in the real-world datasets but are manually constructed from the four well-known datasets. If there are non-IID scenarios in nature that motivate the algorithm, manual construction shouldn't be necessary. Yet, I would not stick to it if this is the common practice in the literature of FL, which seems to be recognized by my fellow reviewers.

---

### Official Review · Reviewer_V6Hn · 2023-11-30

**Soundness:** 2 fair
**Presentation:** 1 poor
**Contribution:** 2 fair
**Rating:** 5
**Confidence:** 3

**Summary:**

This paper proposes a Bayesian framework for clients in a federated setup, which motivates their clustering algorithm. Since, the Bayes optimal assignment is computationally intractable, the paper proposes several heuristics, and empirically compares each with baseline WeCFL on CIFAR10 and FMNIST.

**Strengths:**

1. The authors provide rigorous (albeit hard-to-parse) derivations for a Bayesian framework in clustered federated learning, that can be applied over a large class of priors.
2. The main benefit of the above framework is that it models inter-cluster relationships, and seems to be less sensitive (at least in theory) to the choice of number of clusters K. This can be seen as a more principled form of soft-clustering.
3. Section 3 motivates the need for having approximate association strategies, which the authors investigate in Section 4. The datasets and baselines chosen for the exposition are satisfactory.

**Weaknesses:**

1. Overall, there is a significant concern over the clarity of writing throughout Sections 3 and 4. For example, even though the authors tabulate their notations in Appendix A, it was pretty hard to follow the arguments in Section 3 because of notation overload. Possibly, some of them can be removed, or some of the derivations can be written down by dropping unnecessary indexing wherever possible.
2. Since the different possible associations of clients can be combinatorially very large to evaluate, the authors propose approximations in Section 4. But, it is unclear if BCFL-G/C is helpful in practice, compared to WeCFL. In fact, BCFL-G which does greedy association at each round is quite similar to WeCFL.
3. From point 2, the main contribution I see is BCFL-MH, but this method is computationally heavier given that it requires computing densities in Sec 3.1 and 3.2, and I could not see a computational cost / performance benefit tradeoff to justify BCFL-MH.
4. The authors claim "new theoretical insights" in their introduction, but the paper does not have any formal guarantees for the proposed approximations.
5. The authors do not provide any ablations on the choice of number of clusters K, which can be quite critical for some of the proposed approximations like BCFL-MH.

**Questions:**

1. The derivations in equations 2-7 seem to be simple application of Bayes rule under conditional independence of clients data given model parameters. On the other hand, the approximation in equation 9, which seems to be quite critical is unclear. The authors mention some numerical integration approximation, but it would be useful if they could clarify this further.
2. In Table 1, what are the standard deviations of the reported metrics? Since the difference between WeCFL and BCFL-G/C is quite small, this would help clarify statistical significance of the reported results.
3. The motivation for warmup is unclear. Also, how are the local weights clustered? Is it in the euclidean metric?
4. Have the authors tried using the learnt associations for fine-tuning, as opposed to global training? Maybe there are some general image features that can be learnt from all users?
5. Have the authors run IFCA on their datasets for comparison?

---

> ### Author Response · Authors · 2023-12-02
>
> We thank the reviewer and we reply to the points raised by the reviewer.
>
> Weakness:
>
> 1. We improved the notation readability for those sections, as well as added pseudo-code algorithmic description for both the conceptual solution (section 3) and the practical schemes (section 4).
> We acknowledge the possibility of enhancing these sections for clarity. However, given the constraints of time, it is challenging to undertake substantial revisions in less than a day. Nevertheless, we are committed to addressing the reviewer's concerns to the best of our ability within the time available and in future iterations.
>
> 2. Firstly, the methods we propose are not only for experimental comparison only, but our primary objective is to introduce a framework that can be adapted and extended to accommodate a variety of methods for future research. Secondly, WeCFL represents a non-trivial algorithm, and we do not claim that our methods will universally outperform it. However, as indicated by the results, BCFL-G/C achieve similar or improved performance compared to WeCFL across a diverse range of datasets and experimental configurations, with some additional results presented in the Appendix.
>
> 3. The main contribution is the Bayesian framework for CFL, the specific algorithms like BCFL-MH are options within it.  While BCFL-MH seems to be a good solution performance-wise, the other BCFL-G/Ca approaches provide remarkable performance at lower communication/complexity burdens. We added a discussion on this in the revision, as well as a discussion of the communication and computation cost in Appendix, Section B.
>
> 4. The theoretical insights come from the formulation of the clustering FL problem as a probabilistic model which we address from a Bayesian perspective, dealing with the multiple data associations. This allowed us to propose different practical methods, solidly based on the Bayesian paradigm. We acknowledge the reviewer's concern regarding the absence of a formal guarantee in this part, or at least guarantees in the sense of those provided in frequentist-based methods. As a matter of fact, our proposed approximation BCFL-G has strong connections to the IFCA [ghosh2020efficient], when formulated in a Bayesian context, which provides those theoretical guarantees. Our paper concentrates on the Bayesian framework rather than on particular deterministic algorithms, which is why we did not initially include a convergence proof. However, we recognize the importance of such an analysis, and we are certainly open to investigating convergence guarantees in future research.
>
> 5. We added sensitivity analysis. Please refer to our appendix for results on the selection of $K$ and its justification.

---

> > ### Author Response · Authors · 2023-12-02
> > **Questions**
> >
> > Questions:
> >
> > 1. This is not a mere application of Bayes rule. Of course, as any Bayesian solution, that result is used, but in the derivations pointed out there is much more than this, as testified by the formal notation we provided.
> > We call the reviewer's attention to the fact that the conceptual ideas motivating those derivations are not trivial, and different aspects of the formulation were performed to either, result in a more tractable problem or to connect with the distributed processing paradigm employed in FL.
> >  For instance, the fact that we are not computing the posterior as in Equation (1) but as in Equation (2), allows us to: $(i)$ develop the distributed Bayesian framework which is discussed in Equations (5)-(9); and $(ii)$ allows for a simpler recursive strategy that is presented in Section 3.2.
> >  We highlight that all those steps were carefully thought through and conceptualized to cope with the Bayesian data association and the CFL paradigms, bringing challenges regarding its motivation, design, and conceptualization.
> >  For instance, in Equations (2)-(9) intricate application of Bayes Theorem, marginalization, product rules, and integral approximation were leveraged leading to a posterior formulation that connects with the FL paradigm.
> >  For these reasons, we believe that the derivations presented in Section 3.1, despite their algebraic simplicity, are not trivial and show the chain of thought applied to the problem.
> > Of course, a Bayesian framework will rely on the usage of the Bayesian Theorem in its derivations.
> >  Regarding equation (9), it aims to explain a numerical integration approximation, allowing for effective factorization across local clients. Typically, numerical integration involves sampling; in our implementation approach, we can simplify the process by taking a single sample—specifically, the expected value of the parameter. While Bayesian theory itself is straightforward, the critical aspect lies in its connection to data association theory and its subsequent application to Cluster Federated Learning.
> >
> > 2. We have now included the standard deviation in Appendix, Table 10. Initially, due to the limited space and the small magnitude of the standard deviation, we have omitted these results.
> >
> > 3. Yes, the metric employed in our experiment is the Euclidean metric. We have updated this in the `Baseline Models and System Settings' section of the Appendix. The motivation is just to check if warm-up can speed up the training and improve the results, which indeed did. With the same number of communication rounds, it usually achieves better results. The reason behind this is that we provide a good initial estimate at the beginning, as shown in Figure 3. Warm-up methods typically have good initial accuracy and are used in other works.
> >
> > 4. That's good idea that we would like to explore in future works. There is little time to incorporate this in the present review.
> >
> > 5. Indeed, we implemented the IFCA algorithm and tested in the experiments. Actually, IFCA can be interpreted as the deterministic variant of our greedy BCFL-G method. The results between both were very similar. However, as our study does not focus on contrasting deterministic approaches with Bayesian methods, the comparisons have been excluded from this analysis.

---

### Author Response · Authors · 2023-11-23
**General Response**

We are very thankful to the area chair for coordinating the review of our manuscript and grateful to the reviewers for their valuable feedback.
The authors have provided point-to-point responses to the comments raised by the reviewers. Some changes were made in the revision as blue color, here is the summary of the changes:
- Added a new experiment to characterize the sensitivity to the assumed number of clusters, which is included in the Appendix.
- Figure 3 has been improved by zooming in, making it more clear to interpret.
- Added related work of data association theory to provide more context on the principles that our BCFL framework is based upon. This modeling principle is different to other methods in the CFL literature which are based on consensus approaches. This data association approach enables the practical solutions presented in the paper, while leveraging a principled probabilistic framework based on Bayesian theory.
- Fix other typo errors reviewers point out.

The authors would like to kindly remind the reviewers to take a look at the responses and see whether the raised concerns have been well addressed. Thank you for your help and expertise. We look forward to hearing from you again.
At the same time, we are waiting for another potential reviewer’s feedback, upon which we will update the work accordingly again.

---

### Meta-Review · Area_Chair_V43x · 2023-12-14

**Metareview:**

This paper proposes a Bayesian framework for clustering clients in a federated setup. Based on heuristic approximation versions of their framework they propose clustering/FL algorithms. The paper empirically compares each with baseline WeCFL on CIFAR10 and FMNIST. The general framework is new and can be seen as a bayesian generalization the IFCA framework from [1] that associates the client with a cluster whose model has the lowest loss on the client's data and then uses those associations to update the cluster model. Overall this is promising and well-motivated extension> However all reviewers pointed out a number of concerns that need to be addressed for the paper to make a more clear and convincing case for this approach and none supported accepting the paper. For example (from V6Hn) the main contribution is a more involved variant BCFL-MH, but this method is computationally heavier given that it requires computing densities in Sec 3.1 and 3.2. However a computational cost / performance benefit tradeoff to justify BCFL-MH is missing. While the authors added a discussion during a review process such discussion needed to be present in the submission for a proper evaluation.

[1] Ghosh et al. An efficient framework for clustered federated learning. NeurIPS 2020

**Justification For Why Not Higher Score:**

see above

**Justification For Why Not Lower Score:**

n/a

---

### Decision · Program_Chairs · 2024-01-16

Reject